# Integrating Artificial Intelligence and Precision Therapeutics for Advancing the Diagnosis and Treatment of Age-Related Macular Degeneration

**DOI:** 10.3390/bioengineering12050548

**Published:** 2025-05-20

**Authors:** Mini Han Wang

**Affiliations:** 1Zhuhai People’s Hospital (The Affiliated Hospital of Beijing Institute of Technology, Zhuhai Clinical Medical College of Jinan University), Zhuhai 519000, China; 1155187855@link.cuhk.edu.hk; 2Department of Ophthalmology and Visual Sciences, Chinese University of Hong Kong, Hong Kong 999077, China; 3Zhuhai Institute of Advanced Technology Chinese Academy of Sciences, Zhuhai 519000, China

**Keywords:** age-related macular degeneration, molecular crosstalk, preventive strategies, retinal degeneration, artificial intelligence, bibliographic study, therapeutic strategies

## Abstract

Age-related macular degeneration (AMD) is a multifactorial retinal disease influenced by complex molecular mechanisms, including genetic susceptibility, inflammation, oxidative stress, and metabolic dysregulation. While substantial progress has been made in understanding its pathogenesis, the full molecular underpinnings of AMD remain unclear, impeding the effectiveness of current therapeutic strategies. This study provides an in-depth exploration of the molecular interactions involved in AMD progression, particularly focusing on genetic predispositions (such as CFH, ARMS2/HTRA1, and APOE), inflammatory pathways (including complement system dysregulation and cytokine responses), lipid metabolism (e.g., cholesterol homeostasis and drusen formation), and angiogenesis (VEGF signaling). Through a systematic review and bibliometric analysis of literature published between 2015 and 2025, the study identifies emerging research trends, existing gaps, and promising future therapeutic directions. It further investigates innovative precision medicine approaches, including gene editing (CRISPR), RNA therapeutics (siRNA, antisense oligonucleotides), immunomodulatory therapies, and nanotechnology-based drug delivery systems. Additionally, the study examines the role of metabolic disorders such as diabetes and dyslipidemia in AMD progression, highlighting the influence of systemic health factors on disease onset. Finally, the potential of artificial intelligence (AI) in enhancing AMD management through biomarker-based risk stratification, predictive modeling, and personalized treatment optimization is assessed. By mapping the intricate molecular networks underlying AMD and evaluating novel therapeutic strategies, this research aims to contribute to the development of more effective, individualized treatment protocols for patients with AMD.

## 1. Introduction

Age-related macular degeneration (AMD) [1] is a predominant cause of visual impairment in individuals aged 50 and older, primarily affecting the macula, the central region of the retina responsible for high-resolution vision. AMD is characterized by the progressive degeneration of retinal cells, resulting in the deterioration of central vision while peripheral vision typically remains unaffected [2]. The condition is categorized into two forms: dry AMD, which is associated with the gradual accumulation of drusen, a type of extracellular deposit beneath the retina, and wet AMD, which is characterized by the proliferation of abnormal choroidal blood vessels that leak fluid and contribute to rapid retinal damage. Key risk factors for AMD include advanced age, genetic predisposition, smoking, and hypertension. While a definitive cure remains elusive, early detection and intervention, including pharmacological treatments such as anti-VEGF injections, along with lifestyle adjustments, have been shown to mitigate disease progression and preserve visual function.

The molecular mechanisms underlying retinal degeneration are intricate and involve multiple interconnected pathways, driven by both genetic and environmental factors that disrupt the homeostasis of retinal cells. Central to these processes are oxidative stress, inflammation, apoptosis, and the dysfunction of retinal pigment epithelium (RPE) cells. In conditions such as AMD, the accumulation of reactive oxygen species (ROS) induces oxidative damage to retinal cells, while chronic inflammatory responses exacerbate cellular injury and impair the integrity of the blood–retina barrier. The progressive dysfunction of RPE cells, which are essential for the maintenance and support of retinal neurons, leads to failure in the clearance of metabolic waste products, such as drusen, thereby contributing to the degeneration of photoreceptors and other retinal structures.

At the genetic level, mutations in genes such as complement factor H (CFH) and age-related maculopathy susceptibility 2 (ARMS2) have been identified as key risk factors for retinal degeneration, particularly in the context of AMD. Moreover, the aberrant activation of molecular signaling pathways, such as the complement cascade, has been implicated in the pathogenesis of retinal degeneration, where it mediates inflammation and immune-related damage.

Preventive strategies aimed at mitigating retinal degeneration focus on modulating both the molecular pathways involved and the modifiable risk factors. Pharmacological interventions targeting oxidative stress and inflammation, such as antioxidants and anti-inflammatory agents, are under investigation for their potential to slow disease progression. Additionally, inhibiting specific molecular pathways, such as the complement system in AMD, using complement inhibitors has shown therapeutic potential. Lifestyle modifications, including smoking cessation, adoption of a nutrient-rich diet (exemplified by the AREDS formulation for AMD), and the management of systemic conditions like hypertension, are also critical in reducing the risk of retinal degeneration. Early detection, facilitated by advanced diagnostic imaging technologies, remains essential for timely intervention and the preservation of retinal function.

Scientometrics is an interdisciplinary field that employs quantitative methods to analyze and evaluate scientific literature and research activities. It applies statistical and mathematical techniques to measure various aspects of scientific production, dissemination, and impact. Central to scientometrics is the examination of patterns in scholarly communication, the identification of emerging research trends, and the assessment of the performance of individuals, institutions, and nations within the scientific domain. Commonly used indicators in scientometric analyses include citation metrics, impact factors, the h-index, and bibliometric mapping, which serve to gauge the influence and quality of academic output. The field also encompasses the study of research collaboration networks, funding dynamics, and the role of open access in the global dissemination of knowledge. By integrating elements from information science, sociology of science, and data analytics, scientometrics provides valuable insights into the structure and dynamics of scientific communities, informing decisions in research policy, funding allocation, and academic career development.

This study integrates both bibliometric and qualitative analyses to examine the research landscape of AMD. A bibliometric analysis utilizing the CiteSpace citation exploration tool was performed on 18,356 publications from WoSCC and 19,830 from Scopus, yielding a total of 38,186 unique citations. The analysis identifies the most influential countries, institutions, authors, and citations within the field, while also mapping core journals, emerging research hotspots, and evolving trends. In addition to the bibliometric approach, qualitative analysis is employed to provide a deeper understanding of the molecular mechanisms, preventive and therapeutic strategies, and emerging technological advancements related to AMD. Notably, this study emphasizes the integration of artificial intelligence in the diagnosis and treatment of AMD. The structure of the paper is as follows: the second section outlines the methodology used, while the third section presents the results, followed by a comprehensive discussion. This study offers a holistic overview of the current state of AMD research, highlighting key areas of focus and potential future directions.

As Figure 1 shows, AMD involves complex molecular mechanisms, including oxidative stress, genetic predisposition, complement–lipid interactions, inflammatory pathways, and angiogenesis, all contributing to retinal degeneration. Oxidative stress, driven by an imbalance between reactive oxygen species and antioxidant defenses, leads to cellular damage in the RPE and photoreceptors. Genetic factors, such as mutations in CFH, influence susceptibility to AMD by affecting immune regulation. Complement–lipid interactions exacerbate local inflammation, while inflammatory pathways mediated by cytokines and macrophage recruitment further compromise retinal integrity. In advanced wet AMD, pathological angiogenesis results in neovascularization and vascular leakage, whereas dry AMD is characterized by progressive atrophy of the RPE and photoreceptors [3].

Preventive strategies focus on mitigating these molecular mechanisms through lifestyle and dietary modifications, including diets rich in antioxidants, omega-3 fatty acids, and the use of AREDS supplements. Smoking cessation reduces oxidative stress and inflammation, while UV protection safeguards against phototoxic damage to retinal cells. Therapeutic interventions target specific pathways, including pharmacological agents like anti-VEGF drugs to inhibit angiogenesis and corticosteroids to modulate inflammation. Gene and cell-based therapies aim to restore retinal function through gene editing or stem cell transplantation, while nanomedicine offers advanced drug delivery systems for targeted treatment. Surgical and laser treatments, such as photodynamic therapy, address structural damage and vascular abnormalities. Together, these preventive and therapeutic strategies aim to preserve retinal architecture, slow AMD progression, and improve visual outcomes.

## 2. Materials and Methodology

The methodology of this study combines bibliometric and qualitative analysis to comprehensively examine the research landscape on AMD. Bibliometric analysis utilizes the CiteSpace citation exploration tool to analyze the collected publications. This approach allows for the identification of influential countries, institutions, authors, and key citations, as well as the mapping of core journals, emerging research hotspots, and evolving trends in the field. Qualitative analysis complements the bibliometric approach by focusing on the deeper understanding of the molecular mechanisms, preventive and therapeutic strategies, and technological advancements in AMD research, with particular attention given to the role of artificial intelligence in enhancing diagnostic and treatment strategies. The integration of these two methods enables a multifaceted exploration of AMD research and its future directions.

### 2.1. Data Source

This study compiles literature from two major citation databases: the Web of Science Core Collection (WoSCC) and Scopus, which serve as the primary sources for bibliometric analysis. The search query in WoSCC is configured as: TS = (“age related macular degeneration” OR “AMD”). Similarly, the search query in Scopus is set as: TS = (“age related macular degeneration” OR “AMD”). The time span for the search is from 1 January 2015, to 3 February 2025, with data retrieval conducted on 3 February 2025.

The PRISMA flow diagram illustrates the literature selection process (Figure 2). Records were systematically retrieved through comprehensive database searches conducted in the WoSCC and Scopus. The complete search strategy, including final search strings, Boolean operators, and applied filters (such as language and document type restrictions), is detailed in the Section 2.

In accordance with our grey literature policy, only peer-reviewed publications indexed in WoSCC were considered eligible for inclusion, while non-peer-reviewed sources, such as conference abstracts and unpublished manuscripts, were excluded. Following initial retrieval, duplicate records were identified and removed using CiteSpace software (6.4 R1), with additional manual verification to ensure the accuracy of the de-duplication process.

The remaining records were then subjected to title and abstract screening using pre-established inclusion and exclusion criteria to assess their relevance to the study objectives. Records deemed irrelevant were excluded at this stage. Subsequently, the full texts of potentially eligible articles were reviewed in detail. Articles were excluded if they lacked a specific focus on AMD, failed to meet methodological quality standards, or were not relevant to the scope of the review.

Upon completion of this multi-tiered screening process, a total of 360 peer-reviewed articles met the inclusion criteria and were incorporated into the final analysis. These selected studies form the foundation of this manuscript’s bibliometric evaluation and narrative synthesis. Including a PRISMA-compliant flow diagram and transparent methodological reporting strengthens this review’s reproducibility and scientific rigor.

### 2.2. Analysis Tools

This study employs CiteSpace (6.4.R1, 64-bit) and JRE (Java 17.0.2+8-LTS-86, 64-bit) for bibliometric analysis, both of which can be downloaded from the official CiteSpace website [4]. CiteSpace is an advanced tool designed for visualizing and analyzing scientific literature networks, allowing researchers to examine citation patterns, co-citation relationships, and keyword co-occurrence. It offers a robust approach to bibliometric analysis by enabling users to map the structure of scientific knowledge, identify influential publications, prominent researchers, and emerging research trends. CiteSpace I [5] and CiteSpace II [6], the first and second iterations of the tool, have been cited 3016 and 7427 times, respectively, according to Google Scholar as of 3 February 2025.

In CiteSpace visualizations [7], nodes represent key entities such as articles, authors, or keywords. The shape of the nodes indicates the degree of influence or centrality of an entity, including factors such as the author’s prominence, the citation frequency of a publication, or the significance of a journal, institution, or country. The size of the nodes correlates with their significance, with larger nodes representing higher citation counts or greater prominence in the network. For example, in a co-authorship network, larger nodes may indicate authors with a greater number of publications, while in a keyword co-occurrence network, they may represent terms that appear more frequently across the dataset. Thus, larger node sizes signify higher visibility or relevance within the scholarly domain. Edges (the links between nodes) represent various relationships, such as citations, co-citations, and keyword co-occurrence. Citation links indicate direct citations between articles, co-citation links show articles that are cited together by a third publication, and keyword co-occurrence links highlight thematic overlaps between articles. The weight of the edges reflects the degree of betweenness among nodes, providing insight into the connectivity and structural importance of specific entities within the network.

Centrality scores (e.g., betweenness centrality) are used to quantify the influence or importance of a node (e.g., author, keyword, or institution) within the network, with nodes possessing a centrality value ≥ 0.1 being considered key nodes in the network. A higher centrality score indicates that a node acts as a key connector or bridge in the knowledge structure, often facilitating the flow of information between otherwise disconnected clusters. Moreover, the parameter of network density refers to the proportion of actual connections between nodes relative to the total number of possible connections. It serves as a global measure of network cohesion. A higher density indicates a more interconnected and collaborative network, whereas a lower density suggests fragmentation or specialization across subfields.

In addition to nodes and edges, CiteSpace visualizations often feature annotations and labels, which provide essential details such as article titles, author names, or research topics. Timelines in the visualizations track the evolution of research themes and citation patterns over time, enabling the identification of emerging trends. Clusters, which are formed through clustering algorithms, group related nodes based on shared citations or keywords, and are color-coded to represent different research themes. Furthermore, CiteSpace provides links to external databases and full-text articles, offering researchers a means of deeper exploration into relevant publications. Together, these features make CiteSpace a powerful tool for mapping the structure of scientific literature, uncovering trends, and gaining insights into the development and evolution of scholarly knowledge. In this study, the keyword mapping, historic trend, and cluster classification are displayed. The major authors (according to the centrality), core journals (according to the centrality of publications), major institution (according to the number of publications), most influential counties (according to the centrality of publications), most contributed paper (according to the number of publications), key topics (according to the number of publications) and major involved category (according to the number of publications) are identified.

These metrics provide insights into the structural properties of the research landscape, helping identify influential contributors, emerging topics, and the degree of integration within the scientific community. This study presents an analysis of keyword mapping, historical trends, and cluster classifications derived from bibliometric data. It identifies the major authors, ranked according to centrality, the core journals, determined by the centrality of their published works, and the leading institutions, based on publication volume. Additionally, the most influential countries are highlighted according to the centrality of their publications, while the most contributing papers are selected based on the frequency of their appearances. Key research topics are identified according to the number of associated publications, and the primary research categories are outlined based on publication volume. These analyses provide a comprehensive understanding of the key contributors and trends within the field.

## 3. Bibliographic Analytic Results

### 3.1. Developing Trends in Literature

The historic development of publications related to AMD is illustrated in Figure 3, with a dashed line representing the trend in the number of published studies. The relatively high number of non-duplicate records throughout the period highlights the complementary nature of the WoSCC and Scopus databases, emphasizing the importance of integrating data from multiple sources to access a broader and more diverse range of literature. Non-duplicate publications, representing unique records between the two databases, exhibit a consistent upward trend, starting at 2451 in 2015 and peaking at 4874 in 2024. This reflects a growing body of distinct contributions, indicating an increase in diverse and original research outputs. Between 2016 and 2023, the publication counts from WoSCC and Scopus are relatively similar, suggesting that both databases comprehensively capture AMD-related research, though WoSCC shows slightly higher counts in earlier years and Scopus slightly exceeds WoSCC in later years. A notable rise in publications occurred in 2020, with non-duplicate records increasing to 3775 from 3299 in 2019, likely reflecting enhanced research activity during the COVID-19 pandemic [8]. The upward trend continued into 2021 and 2022, with 2024 emerging as the peak year for AMD research, achieving the highest number of publications across all categories (WoSCC: 2246; Scopus: 2628; non-duplicate: 4874). This peak indicates a culmination of efforts and a growing focus on addressing challenges associated with AMD.

### 3.2. Analysis of Countries and Institutions

Figure 4 and Table 1 show, the country distribution analysis generated by CiteSpace provides a comprehensive overview of global contributions to the research field, highlighting the distribution of publications across countries, ranked by contribution count, centrality scores, and year of publication. The configuration parameters-index (k = 25), LRF = 3.0, L/N = 10, LBY = 5, and e = 1.0—are tailored to prioritize influential nodes and meaningful connections, ensuring a focus on highly cited publications and contemporary trends within the past five years. The network consists of 337 nodes (countries) and 2462 edges (collaborative or citation relationships), with a sparse density of 0.0435, indicative of strong intra-cluster collaborations but less frequent global interactions. The analysis underscores the leading roles of countries such as the United States, China, and Germany, which dominate in both publication counts and centrality scores, reflecting their significant influence and interconnectedness within the global research landscape. Notably, the United States has the highest contribution, with 6721 publications in 2015 and a centrality score of 0.08 when combined with “UNITED STATES”, underscoring its pivotal role in global collaboration networks [9]. China and Germany also demonstrate strong contributions, with centrality scores of 0.04 and 0.06, respectively, highlighting their growing importance in international research.

Countries with high centrality scores, such as Canada (0.11), Switzerland (0.06), and the United Kingdom (0.05), serve as key hubs for global research interactions, connecting diverse scientific networks. Emerging economies, including Saudi Arabia (0.05) and Mexico (0.04), are actively contributing to research while gradually strengthening their international ties. European countries, such as France and Italy, exhibit consistent centrality, underscoring their interconnectedness and collaborative efforts. Asian countries, including Japan and South Korea, display high publication counts but relatively lower centrality, suggesting a focus on regional research with potential for broader global engagement. Specialized contributors like Costa Rica and Georgia exhibit high centrality relative to their publication volumes, indicating their importance in connecting niche research areas globally. Conversely, countries like Turkey, Portugal, and Russia demonstrate moderate contributions with lower centrality, reflecting a focus on domestic or regional research activities. These findings highlight the complex dynamics of global research contributions and collaborations, emphasizing the roles of both established and emerging research nations in shaping the field.

As Figure 5 and Table 2 show, the institution distribution analysis generated by CiteSpace, using criteria such as g-index (k = 25), LRF = 3.0, L/N = 10, LBY = 5, and e = 1.0, offers a comprehensive overview of global research contributions. The network comprises 827 nodes (institutions) and 6959 edges (collaborative or citation relationships), with a density of 0.0204, signifying a sparse structure dominated by strong intra-cluster collaborations but limited global connectivity. Leading institutions, including the University of Melbourne (430) and the University of California (409), are central to advancing the field. North American and European institutions, such as Harvard Medical School, University College London, and the University of Bonn, consistently exhibit high research outputs, demonstrating their dominance in the academic landscape. At the same time, emerging contributors like Sun Yat-Sen University (227) and Shanghai Jiao Tong University (193) highlight Asia’s growing presence in the research community.

Institutions such as the National Eye Institute (NEI) (392) and Singapore Eye Research Institute (247) play pivotal roles in fostering collaborations, despite having lower publication counts compared to leading institutions. Additionally, specialized institutions like Moorfields Eye Hospital (217) and the Cleveland Clinic (209) exhibit consistent influence within their domains, contributing to advancements in niche areas. The emergence of newer players, such as Shanghai Key Lab Ocular Fundus Disease (2022) and Tsinghua University (2023), reflects a dynamic expansion in research domains. This diversification underscores the increasing global contributions from Asian and European institutions, such as Shanghai Jiao Tong University School of Medicine and the University of Eastern Finland, signaling a shift toward a more balanced and interconnected academic ecosystem.

The institutional clustering analysis (Figure 5) highlights the collaborative networks and significant contributions of leading organizations in the research of AMD and related conditions. Prominent institutions such as the University of California, Los Angeles (UCLA), the University Vita-Salute San Raffaele, and the National Eye Center emerge as central hubs driving research efforts, particularly in clinical trials and innovative therapeutic approaches. Cluster #0 (Related Macular Degeneration) and Cluster #2 (Age-Related Macular Degeneration Treatments Trial) underscore the focus of these institutions on evaluating treatment efficacy and advancing therapeutic interventions. Additionally, Cluster #6 (Population-Based Cohort Study) reflects the contributions of institutions conducting large-scale epidemiological research to investigate demographic and genetic risk factors associated with AMD. Cluster #1 (Geographic Atrophy) highlights institutions that focus on the pathophysiological and progressive aspects of advanced AMD. Similarly, Cluster #3 (Choroidal Vasculopathy) and Cluster #4 (Retinal Blindness) reveal efforts concentrated on exploring vascular and retinal changes linked to AMD. The Systematic Review Cluster (#7), prominently featuring contributions from the University of Copenhagen, underscores collaborative endeavors to synthesize and consolidate knowledge, providing evidence-based insights for clinical and research advancements. Collectively, the clustering analysis underscores the interdisciplinary and global nature of AMD research, with key academic and clinical institutions playing pivotal roles in advancing the understanding, diagnosis, and treatment of this complex condition.

### 3.3. Analysis of Cited Journals, Cited Authors, and Cited References

The analysis of the top 20 ophthalmology journals based on publication numbers (Figure 6 and Table 3), as of 6 February 2025, reveals a strong concentration of research in several high-impact journals. Investigative Ophthalmology & Visual Science leads with 18.43% of total publications, followed by Retina and Ophthalmology, with 3.39% and 3.10%, respectively. Journals like Ophthalmology (8.462 impact factor) and American Journal of Ophthalmology (5.711 impact factor) stand out for their significant contributions to clinical and experimental research, while specialized journals like Graefe’s Archive for Clinical and Experimental Ophthalmology and Experimental Eye Research continue to advance the understanding of ocular diseases. PLOS ONE and Scientific Reports reflect the broader, interdisciplinary interest in ophthalmology, while JAMA Ophthalmology and International Journal of Molecular Sciences highlight the integration of ophthalmology with fields like molecular biology and systemic diseases, maintaining high impact factors of 7.492 and 5.923, respectively. These data underscore the importance of both specialized and high-visibility journals in advancing ophthalmic research across various subfields.

The CiteSpace analysis of author distribution in the field (Table 4) reveals a moderate collaborative structure, characterized by isolated research clusters centered around specific institutions and research consortia. The network density (0.009) indicates limited global collaboration, with authors tending to work within smaller, localized groups rather than forming extensive international networks. Prolific contributors such as Francesco Bandello (310 publications) and Ursula Schmidt-Erfurth (305 publications) dominate the field, with their research often anchored in leading institutions like the San Raffaele Scientific Institute and Queen’s University of Belfast. The peak publication period for many of these authors, between 2015 and 2020, reflects significant advancements in imaging technologies, funding initiatives, and collaborative research in ophthalmology. Collaborative hubs like the Bascom Palmer Eye Institute, L.V. Prasad Eye Institute, and Queen’s University of Belfast have played central roles in driving research output and fostering localized partnerships.

As Table 4 shows, emerging contributors, including Enrico Borrelli, Riccardo Sacconi, and Tiarnan D. L. Keenan, signal a shift toward new research frontiers, particularly in the integration of artificial intelligence, deep learning, and big data within ophthalmology. These authors represent growing areas of innovation, such as advanced imaging, retinal disease therapeutics, and precision diagnostics. Co-authorship clusters around figures like Giuseppe Querques [10] and Usha Chakravarthy [2] highlight the collaborative nature of large-scale studies and consortia-based research. Furthermore, the peak productivity years for many researchers align with significant technological breakthroughs and advancements in imaging methodologies. The field’s progression from 2015 onward underscores the increasing influence of emerging technologies and international collaborations in advancing ophthalmic research.

**Table 4 bioengineering-12-00548-t004:** Top 5 authors are listed according to centrality.

Ranking	Authors	Count	Institution	Involved Publications
1	Francesco Bandello	310	IRCCS San Raffaele Scientific Institute, Milan, Italy	Macular Atrophy in Neovascular Age-Related Macular Degeneration: A Systematic Review and Meta-Analysis [11].
2	Ursula Schmidt-Erfurth	305	Medical University of Vienna, Vienna, Austria	Features of intermediate and late dry age-related macular degeneration on adaptive optics ophthalmoscopy: Pinnacle Study Report 8 [11].
3	Giuseppe Querques	278	School of Medicine, Vita-Salute San Raffaele University, Milan, Italy	Delineating the contours of PVAC and TelCaps [12].
4	Usha Chakravarthy	259	Queen’s University of Belfast, Belfast, United Kingdom	Interplay between Lipids and Complement Proteins—How Multiomics Data Integration Can Help Unravel Age-related Macular Degeneration Pathophysiology: A Proof-of-concept Study [13].
5	Frank G. Holz	233	0	Heterogenous visual function deficits in intermediate age-related macular degeneration—A MACUSTAR report [14].

The author clustering visualization generated by CiteSpace (Figure 6) highlights the collaborative landscape and research focus among authors contributing to the field of ophthalmology, particularly AMD and related conditions. The central clusters, such as Cluster #1 (Optical Coherence Tomography Angiography) and Cluster #4 (Pinnacle Study Report), represent dense networks of researchers working on advanced imaging technologies and pivotal clinical trials. These clusters underscore the emphasis on leveraging imaging techniques like optical coherence tomography angiography (OCTA) to explore retinal pathologies and evaluate treatment efficacy. Similarly, Cluster #6 (Eye-Risk Consortium) indicates a significant focus on population-based studies, reflecting collaborative efforts to identify risk factors and genetic predispositions for AMD.

Peripheral clusters, Cluster #15 (Oxidative Stress) and Cluster #16 (HMGA2 mRNA), provide insight into molecular and genetic mechanisms underlying retinal degeneration. These clusters suggest that oxidative stress and gene expression studies are integral to understanding disease progression and developing targeted therapies. Clusters like Cluster #18 (Tissue Inhibitor) further highlight the exploration of angiogenesis inhibitors as therapeutic interventions. Overall, the distribution of clusters illustrates a dynamic interplay between clinical applications, molecular research, and emerging technologies, such as artificial intelligence and deep learning, which are poised to transform the field. This comprehensive collaboration and research network emphasizes the multidisciplinary nature of ophthalmic research, addressing clinical, technological, and molecular challenges.

The most influential studies (Table 5, retrieval time is 6 February 2025) show that the field of ophthalmology has seen substantial progress in the treatment and comprehension of AMD, which remains one of the leading causes of blindness globally. One of the most influential studies in this regard is Schwartz et al. (2015) [15], which evaluated the safety and potential of human embryonic stem cell-derived retinal pigment epithelium (hESC-RPE) transplants for treating Stargardt’s macular dystrophy and AMD. This study demonstrated that hESC-RPE transplants were safe, with no adverse effects like proliferation or rejection. The visual acuity of many patients improved or remained stable, indicating the potential for regenerative stem cell therapies in retinal diseases. This research highlighted the viability of stem cell-based approaches to enhance visual outcomes and quality of life for patients suffering from retinal degenerative diseases.

Another critical area of research is the role of oxidative stress and inflammation in AMD pathogenesis. The review by Loboda et al. (2016) [16] examined the Nrf2/HO-1 system and its role in mitigating oxidative damage, which is a key factor in diseases like AMD. This pathway’s cytoprotective effects, particularly its involvement in antioxidant defense and stress protection, suggest that targeting this mechanism could slow down retinal cell degeneration in patients with AMD. The study emphasized the importance of oxidative stress in age-related diseases and positioned Nrf2/HO-1 modulation as a promising therapeutic target to counteract the progression of AMD.

Furthermore, Kermany et al. (2018) [17] conducted a meta-analysis of global blindness and vision impairment, identifying AMD as one of the major contributors. The study projected a significant rise in AMD cases globally, underscoring the need for comprehensive eye care strategies, particularly in aging populations. This global health perspective called for improvements in early detection, diagnosis, and treatment access to prevent vision loss from AMD, which is expected to increase substantially as the global population ages.

In addition to the growing understanding of disease mechanisms, technological advancements have played a significant role in diagnosing AMD. Kermany et al. (2018) [17] also explored the potential of artificial intelligence (AI) and deep learning [18] in diagnosing macular degeneration and diabetic retinopathy from medical images, such as optical coherence tomography (OCT). This AI-driven approach demonstrated accuracy comparable to human experts, highlighting the potential for AI in clinical decision support systems. These findings suggest that AI has the capacity to revolutionize medical imaging and diagnostics, offering an efficient and scalable solution, particularly for regions with limited access to specialists. However, the study also pointed out the challenges of implementing AI in clinical practice, such as algorithm explainability and concerns regarding the acceptance of AI systems by both physicians and patients.

The clinical treatment landscape for AMD has also evolved, as evidenced by Martin et al. (2020) [19], who compared the effects of ranibizumab and bevacizumab for treating neovascular AMD. The study showed that both drugs were similarly effective in improving visual acuity, but monthly treatment resulted in greater benefits compared to as-needed treatment. Furthermore, the study highlighted that bevacizumab was associated with a higher rate of systemic adverse events compared to ranibizumab. These findings provided valuable insights into treatment protocols, emphasizing the importance of personalized treatment regimens that balance efficacy with patient safety.

Finally, Steinmetz et al. (2021) [20] conducted an analysis of global blindness and vision impairment trends, revealing that while there had been a reduction in avoidable blindness, the aging population and issues like uncorrected refractive errors continued to contribute significantly to the global burden of vision impairment. The study emphasized the need for improved preventive measures, particularly in addressing cataract surgery access and refractive error correction, while also acknowledging the growing impact of AMD. The findings called for continued efforts to address avoidable blindness through public health initiatives, while also recognizing the increasing prevalence of AMD as a key challenge in vision care.

Thus, these studies collectively reflect the dynamic and evolving landscape of AMD research and treatment. From stem cell therapies offering regenerative potential to AI enhancing diagnostic accuracy, the field is making strides in both clinical and technological domains. However, significant challenges remain in translating these innovations into widespread clinical practice, particularly concerning accessibility, affordability, and the ethical implications of emerging technologies. As research continues, it will be crucial to balance scientific progress with practical solutions that can reach diverse populations and improve patient outcomes globally.

According to the analysis in Table 5, the most highly cited journals in the field, reflecting their significant impact on the research landscape of AMD, include The Lancet (Impact Factor: 2023—91.245), Cellular Molecular Life Sciences (Impact Factor: 2023—6.662), The Lancet Global Health (Impact Factor: 2023—20.199), Cell (Impact Factor: 2023—38.641), and British Journal of Ophthalmology (Impact Factor: 2023—2.601). These publications are pivotal in advancing knowledge in ophthalmology, particularly in AMD. Other prominent journals, such as Ophthalmology (Impact Factor: 2023—8.462) and JAMA (Impact Factor: 2023—7.492), contribute extensively to the dissemination of groundbreaking research on ocular diseases and their management. Journals like Signal Transduction and Targeted Therapy (Impact Factor: 2023—7.235) and Aging Disease (Impact Factor: 2023—6.831) further enhance the understanding of the molecular mechanisms of AMD, offering critical insights into therapeutic approaches and the biological processes underpinning the disease. Collectively, these journals represent a vital source of high-impact research, shaping the future directions of AMD-related science and clinical practice.

**Table 5 bioengineering-12-00548-t005:** The most cited references (2015–2025) are listed according to the number of being cited.

Year	Reference	Findings	Hotspot	Journal	Number of Cited by (According to Google Scholar)
2015	Human embryonic stem cell-derived retinal pigment epithelium in patients with age-related macular degeneration and Stargardt’s macular dystrophy: follow-up of two open-label phase 1/2 studies [15]	This study provides evidence of the medium- to long-term safety of human embryonic stem cell (hESC)-derived retinal pigment epithelium transplants for treating Stargardt’s macular dystrophy and age-related macular degeneration. No adverse proliferation or rejection was observed, and visual acuity improved or remained stable in many patients, indicating potential for hESC-derived cells in regenerative treatments.	Human Embryonic Stem Cells (hESC), Retinal Pigment Epithelium, Regenerative Medicine, Stargardt’s Macular Dystrophy, Age-related Macular Degeneration, Stem Cell Transplantation, Visual Acuity, Safety, Graft Survival, Tissue Repair, Immune Rejection, Subretinal Transplantation, Phase 1/2 Studies, Quality of Life	The Lancet	1530
2016	Role of Nrf2/HO-1 system in development, oxidative stress response and diseases: an evolutionarily conserved mechanism [16]	This review explores the multifaceted roles of Nrf2 and its target heme oxygenase-1 (HO-1), emphasizing their cytoprotective, anti-aging, and stress-protective effects. Disturbances in HO-1 levels are linked to age-related disorders like neurodegeneration, cancer, and macular degeneration, suggesting a conserved function across species.	Nrf2, Heme Oxygenase-1 (HO-1), Antioxidant, Anti-inflammatory, Detoxification, Aging, Neurodegeneration, Macular Degeneration, Stress Protection, Longevity, Transcription Factors, Cytoprotection, Apoptosis, Angiogenesis, Age-dependent Disorders	Cellular molecular life sciences	2339
2017	Global causes of blindness and distance vision impairment 1990–2020: a systematic review and meta-analysis [17]	This systematic review and meta-analysis identified major causes of vision impairment and blindness globally, with uncorrected refractive error and cataracts as the leading contributors. The study also projected a significant rise in cases by 2020, underscoring the need for increased eye care to address preventable vision loss, particularly in adults aged 50 and older.	Vision Impairment, Blindness, Uncorrected Refractive Error, Cataract, Age-related Macular Degeneration, Glaucoma, Diabetic Retinopathy, Population-based Studies, Eye Care, Preventable Vision Loss, Systematic Review, Meta-analysis, Global Health	The Lancet Global Health	3499
2018	Identifying medical diagnoses and treatable diseases by image-based deep learning [17]	This research developed an AI system using transfer learning to classify macular degeneration, diabetic retinopathy, and pediatric pneumonia from medical images. The system demonstrated comparable accuracy to human experts and provided interpretable diagnoses, with potential for broad applications in biomedical imaging.	AI, Transfer Learning, Macular Degeneration, Diabetic Retinopathy, Pneumonia Diagnosis, OCT, Chest X-rays, Deep Learning, Clinical Decision Support, Medical Imaging	Cell	3929
2019	Artificial intelligence and deep learning in ophthalmology [21]	Deep learning has shown significant potential in ophthalmology, especially in detecting major eye diseases like diabetic retinopathy, glaucoma, macular oedema, and age-related macular degeneration. However, challenges remain in clinical deployment, including algorithm explainability, medicolegal issues, and acceptance from both physicians and patients.	Deep Learning, Ophthalmology, AI, Image Recognition, Diabetic Retinopathy, Glaucoma, Macular Oedema, Age-related Macular Degeneration, Telemedicine, Algorithm Explainability, Clinical Challenges, Medicolegal Issues, Patient and Physician Acceptance, Ocular Imaging	British Journal of Ophthalmology	1247
2020	Ranibizumab and Bevacizumab for Treatment of Neovascular Age-related Macular Degeneration Two-Year Results [19]	Ranibizumab and bevacizumab had similar effects on visual acuity over two years, with monthly treatment resulting in greater improvements than as-needed treatment. Switching from monthly to as-needed treatment led to decreased visual acuity and a lower proportion of patients without fluid, while bevacizumab was associated with a higher rate of systemic serious adverse events compared to ranibizumab.	Ranibizumab, Bevacizumab, Neovascular Age-related Macular Degeneration, Visual Acuity, Monthly Treatment, As-needed Treatment, Clinical Trial, Systemic Adverse Events, Vascular Endothelial Growth Factor, Treatment Regimen, Drug Comparison, Fluid Resolution, Arteriothrombotic Events, Serious Adverse Events	Ophthalmology	It could not find the citation on Google Scholar, according to Web of Science, the citation is 1142.
2021	Causes of blindness and vision impairment in 2020 and trends over 30 years, and prevalence of avoidable blindness in relation to VISION 2020: the Right to Sight: an analysis for the Global Burden of Disease Study [20]	The study found that, despite a reduction in age-standardized rates of avoidable blindness, the target set by the World Health Assembly Global Action Plan was not achieved in the global reduction in avoidable vision impairment (MSVI). The prevalence of avoidable blindness and MSVI increased due to the ageing population, with cataract and undercorrected refractive error being the leading causes of both blindness and MSVI in adults aged 50 and older.	Avoidable Vision Impairment, Blindness, Cataract, Undercorrected Refractive Error, Age-related Macular Degeneration, Diabetic Retinopathy, Glaucoma, Global Action Plan (WHA GAP), Age-standardized Prevalence, Eye Health Services, Systematic Review, Meta-analysis, Ageing Population, Public Health, MSVI	The Lancet Global Health	1751
2022	Efficacy, durability, and safety of intravitreal faricimab up to every 16 weeks for neovascular age-related macular degeneration (TENAYA and LUCERNE): two randomized, double-masked, phase 3, non-inferiority trials [22]	In two phase 3 trials (TENAYA and LUCERNE), intravitreal faricimab (6.0 mg) administered up to every 16 weeks was shown to be non-inferior to aflibercept (2.0 mg every 8 weeks) in improving best-corrected visual acuity in patients with neovascular age-related macular degeneration. Rates of ocular adverse events were similar for both treatments.	Faricimab, Neovascular Age-Related Macular Degeneration (nAMD), Intravitreal Injections, Best-Corrected Visual Acuity, Phase 3 Trials, Aflibercept, Ocular Adverse Events, Non-Inferiority, Angiopoietin-2, Vascular Endothelial Growth Factor A (VEGF-A), Treatment Regimens, TENAYA, LUCERNE.	The Lancet	445
2023	Endoplasmic reticulum stress: molecular mechanism and therapeutic targets [23]	This review highlights the pivotal role of endoplasmic reticulum stress in various ocular diseases, including glaucoma, diabetic retinopathy, age-related macular degeneration, and cataracts, among others. It discusses how ER stress disrupts proteostasis and triggers adaptive responses that impact transcription and protein processing. Additionally, it examines therapeutic strategies aimed at alleviating ER stress, such as drugs, gene therapy, and stem cell therapy, as potential treatments for these diseases.	Endoplasmic Reticulum (ER), ER Stress, Proteostasis, Ocular Diseases, Glaucoma, Diabetic Retinopathy, Age-Related Macular Degeneration, Retinitis Pigmentosa, Protein Homeostasis, Unfolded Protein Response, Therapeutic Strategies, Gene Therapy, Stem Cell Therapy, Neurotrophic Factors, Autophagy, Ocular Tumors, Myopia.	Signal transduction targeted therapy	
2024	Age-Related Macular Degeneration A Review [2]	AMD is a major cause of vision impairment in older adults, with increasing global prevalence. It is primarily influenced by genetic and environmental factors like smoking, and its progression is characterized by either geographic atrophy or neovascularization. Treatment with anti-VEGF injections and nutritional supplements has been shown to slow progression and improve visual outcomes in patients with AMD.	AMD, Genetic Factors, Smoking, Geographic Atrophy, Neovascular AMD, Anti-VEGF Injections, Visual Acuity, Optical Coherence Tomography, Nutritional Supplements, Vitamin C, Vitamin E, Zinc, Carotenoids, Heritability, Incidence. 4o	JAMA	104
2025	Recent Advances in Our Understanding of Age-Related Macular Degeneration: Mitochondrial Dysfunction, Redox Signaling, and the Complement System [24]	AMD is a multifactorial retinal disease driven by aging, oxidative stress, inflammation, vascular dysfunction, and complement activation, with a significant global impact. While current treatments target neovascular AMD, recent approvals of therapies for geographic atrophy offer hope. This review highlights the role of oxidative stress, mitochondrial dynamics, and the complement system in AMD pathophysiology and discusses novel therapeutic approaches and emerging treatments.	AMD, Geographic Atrophy (GA), Oxidative Stress, Mitochondrial Dynamics, Complement System, Aging, Inflammation, Redox Status, Choroidal Neovascularization (CNV), Emerging Therapies, Drug Therapies, Retinal Disease, Cell Death Pathways.	Aging disease	13

### 3.4. Analysis of Categories, Hotspots, and Burst Topic Historic Trends

The category distribution (Figure 7) delivers a comprehensive overview of the key research areas and their interconnections within a given scholarly corpus. The dataset reveals a significant concentration of research in fields including Ophthalmology, Biochemistry & Molecular Biology, Pharmacology & Pharmacy, and Medicine, Research & Experimental, suggesting that these domains are central to current scientific inquiry, particularly in relation to AMD and other ophthalmic conditions. The high frequency and centrality of these categories indicate their pivotal role in the ongoing academic discourse, with a notable emphasis on both medical and pharmaceutical research in eye diseases.

The centrality metric (shown in Table 6) highlights the significance of specific categories within the broader research network. Fields like Ophthalmology and Biochemistry & Molecular Biology exhibit elevated centrality scores, underscoring their critical position in the interdisciplinary exchange of knowledge. This is further evidenced by the substantial volume of publications within these fields, which reflects a growing body of research focused on the biological mechanisms, pathophysiology, and treatment modalities for degenerative eye diseases, such as AMD.

Furthermore, the inclusion of Neurosciences, Biomedical Engineering, and Immunology in the category distribution points to an increasing integration of biological research with technological advancements. The application of imaging technologies, including OCT, as well as the incorporation of AI into diagnostic practices, is driving innovation in the understanding and management of retinal diseases. These trends are further corroborated by the strong presence of Biotechnology and Engineering categories, reflecting the technological evolution in medical research and clinical applications.

The keyword distribution (Table 7) generated by CiteSpace reflects a comprehensive analysis of frequently occurring terms, providing insight into the primary focus areas and research trends. The most prominent keywords, such as “human”, “age-related macular degeneration”, “humans”, “article”, and “macular degeneration”, highlight the predominant themes within the dataset, focusing on human-centered studies and age-related eye diseases [25]. This distribution reveals that academic interest is strongly concentrated on understanding macular degeneration, its related pathologies, and the methodologies to study them. Additionally, keywords like “male”, “female”, and “aged” indicate a demographic focus, while “optical coherence tomography” and “controlled study” signify a reliance on advanced imaging techniques and structured methodologies.

Furthermore, the presence of terms such as “visual acuity”, “retinal pigment epithelium”, “ranibizumab”, and “diabetic retinopathy” emphasizes the intersection of clinical treatments and pathological conditions. Other terms like “genetics”, “angiogenesis inhibitors”, and “oxidative stress” suggest a deeper exploration of molecular mechanisms and therapeutic interventions. The inclusion of emerging technologies, such as “deep learning” and “optical coherence tomography angiography”, indicates a growing adoption of artificial intelligence and advanced imaging in ophthalmology. Overall, the keyword distribution provides a snapshot of research priorities, with a balance between clinical applications, demographic considerations, and innovative technologies. This serves as a basis for understanding the evolution of the field and identifying potential areas for future research.

The cluster visualization (Figure 8) based on keywords offers a detailed depiction of the evolving landscape of research on AMD and its associated conditions, spanning from 2015 to 2025. The timeline reveals a shift in focus across several key thematic clusters, signifying both a deeper understanding of AMD and the increasing incorporation of advanced technologies into ophthalmic research.

From 2015 to 2020, AMD remains a central theme, consistently positioned at the core of the network, highlighting its ongoing prominence in the field. Early research focused on biological mechanisms and pathological processes, with terms such as choroidal neovascularization, oxidative stress, and retinal pigment epithelium emerging as pivotal areas of study. Additionally, OCT and related imaging technologies are strongly represented, reflecting their critical role in advancing the diagnosis and management of AMD.

In the latter part of the period (2020–2025), AI became a dominant force in the field. Keywords like artificial intelligence model, deep learning, machine learning, and convolutional neural network (CNN) reflect the growing integration of AI in the research landscape. This shift is indicative of the significant role that computational models are playing in improving diagnostic accuracy, treatment planning, and predictive capabilities in AMD management. Furthermore, the increasing presence of terms such as geographic atrophy, neovascular age-related macular degeneration, and subretinal fibrosis points to a growing focus on specific subtypes of AMD and the exploration of novel therapeutic targets.

As the timeline progresses into 2021 and beyond, emerging concepts such as factor H, macular pigment, and geographic atrophy gain prominence, suggesting a shift towards understanding the genetic, systemic, and environmental factors that influence AMD. Additionally, the terms dry AMD and wet AMD show notable bursts in the later years, underscoring the continued emphasis on understanding the differences between AMD subtypes and their distinct treatment approaches.

The cluster relationships highlighted in the visualization also underscore the interdisciplinary nature of AMD research, illustrating the interplay between biological mechanisms, clinical features, and technological advancements. The connections between terms such as macular degeneration, retinal degeneration, diabetic retinopathy, and neovascular age-related macular degeneration point to the growing integration of multiple research domains, particularly in the areas of genetics, pathology, and clinical diagnostics. The prominence of machine learning and deep learning further emphasizes the increasing reliance on computational tools for analyzing complex datasets, enabling more accurate diagnoses and better-informed treatment decisions.

Furthermore, the visualization suggests that the next phase of AMD research will likely center around refining AI-driven models for early detection, treatment prediction, and personalized care. The increased focus on genetic factors, as reflected in terms such as geographic atrophy and factor H, indicates a shift towards targeted therapies and a more individualized approach to managing AMD. Moreover, the continued development of multimodal imaging combined with AI tools will likely enhance the precision and effectiveness of diagnostic practices, providing valuable insights into disease progression and treatment outcomes.

Thus, the keyword and cluster visualization capture the dynamic nature of AMD research, illustrating both the evolving scientific understanding of the disease and the transformative impact of technological innovations such as AI. This progression reflects a broader trend towards integrating computational tools with clinical practice, offering promising avenues for improving AMD diagnosis, prognosis, and treatment in the coming years.

The citation burst (Figure 9) analysis reveals a dynamic progression in the research landscape surrounding AMD and related ophthalmic conditions. From 2015 to 2020, terms such as choroidal neovascularization and vascular endothelial growth factor experienced significant citation bursts, highlighting their central role in retinal and AMD research, particularly in understanding vascular mechanisms driving these conditions. The terms macular degeneration and retinal pigment epithelium also emerged as focal points, reflecting the foundational studies on the biological underpinnings of macular degeneration.

A notable shift occurred with the integration of advanced technologies into ophthalmic research. The inclusion of OCT and spectral-domain optical coherence tomography from 2015 to 2019 marked the rising importance of imaging technologies in the diagnosis and management of retinal diseases, particularly AMD. Moreover, the substantial rise of deep learning from 2017 onward, continuing through 2025, underscores the increasing reliance on AI for disease detection and management. Similarly, terms like machine learning and artificial intelligence have gained significant traction since 2021, reflecting their growing role in enhancing diagnostic accuracy and predictive capabilities in ophthalmology.

More recent trends indicate a broadening of the research scope. Terms such as intraocular inflammation, diabetic retinopathy, and neovascular age-related macular degeneration began gaining momentum in 2021, emphasizing the multifaceted nature of AMD research, which now also addresses inflammation and its contribution to retinal diseases. Additionally, age-related macular degeneration continues to show strong citation bursts through 2025, signaling sustained research interest in this area.

Emerging topics such as geographic atrophy and Mendelian randomization are gaining prominence, marking a shift toward understanding genetic and systemic factors contributing to AMD. These topics, with citation bursts extending into 2022–2025, suggest that future research will increasingly focus on the genetic epidemiology of AMD and other retinal conditions.

In conclusion, the citation burst analysis reveals a clear trajectory in AMD research, from foundational studies in retinal biology and imaging technologies to the current integration of AI and genetic research. This evolving landscape underscores the interdisciplinary nature of ophthalmic research, with innovations in technology and AI playing a pivotal role in advancing the understanding and management of AMD and related diseases.

## 4. Qualitative Analytic Result

### 4.1. Molecular Mechanisms of AMD

AMD is a multifactorial disease influenced by genetic susceptibility, oxidative stress, lipid dysregulation, chronic inflammation, and angiogenic processes. These interconnected molecular mechanisms contribute to disease progression and visual deterioration. A comprehensive understanding of these underlying pathways is essential for developing targeted therapeutic strategies and identifying early diagnostic biomarkers to mitigate AMD-related vision loss.

#### 4.1.1. Genetic Contributions to AMD Pathogenesis

Genetic predisposition is a key determinant in the pathogenesis of AMD, with multiple high-risk genetic variants identified through genome-wide association studies (GWAS). Among these, CFH and ARMS2 have been extensively investigated for their role in AMD progression. Mutations in CFH impair alternative complement pathway regulation, resulting in chronic immune dysregulation, sustained inflammation, and complement overactivation, which collectively contribute to RPE dysfunction, drusen accumulation, and progression toward GA in dry AMD. In contrast, ARMS2 polymorphisms have been implicated in oxidative stress responses and extracellular matrix (ECM) remodeling, potentially increasing susceptibility to CNV in wet AMD. Recent population-based research further supports the link between genetic risk scores (GRS) and AMD severity, demonstrating associations between higher complement and ECM-specific genetic risk and the presence of high-risk intermediate AMD features, including reticular pseudodrusen (RPD) and large drusen area (LDA). Furthermore, emerging multiomics studies highlight the interplay between complement activation and lipid metabolism dysfunction in AMD, suggesting a more intricate molecular landscape than previously understood. These findings underscore the potential clinical utility of genetic risk stratification in AMD, which could enhance early detection, risk prediction, and personalized intervention strategies [26].

#### 4.1.2. Complement–Lipid Interactions in AMD: Insights from Multiomics Studies

Recent multiomics studies have uncovered a significant correlation between complement system activation and lipid metabolism dysregulation in AMD pathogenesis [13]. Two distinct complement–lipid clusters have been identified, offering a deeper understanding of the molecular interplay influencing disease progression. Cluster 1 includes mannan-binding lectin serine protease 1 (MASP1) and high-density lipoprotein (HDL) particles, suggesting a role in immune regulation and lipid transport, whereas Cluster 2 comprises complement factor H-related protein 1 (CFHR1), carboxypeptidase N subunit 2 (CPN2), and complement component C8 gamma chain, along with sphingomyelin and cholesterol variants, which are implicated in cell membrane integrity and inflammatory processes. Additionally, reduced levels of complement protein 1R and sphingomyelin have been associated with a twofold increased risk of AMD (Odds Ratio = 2.13 [1.09, 4.17]), indicating that AMD is driven by a complex interaction between immune dysregulation and lipid-mediated oxidative damage [13]. These findings emphasize the importance of multi-target therapeutic interventions to address the diverse pathological mechanisms underlying AMD.

#### 4.1.3. Oxidative Stress and Retinal Pigment Epithelium Degeneration

Oxidative stress is a major contributor to AMD pathogenesis, particularly in dry AMD, where chronic exposure to ROS leads to cumulative retinal damage. The RPE, responsible for maintaining photoreceptor function and clearing metabolic waste, is highly susceptible to oxidative stress-induced dysfunction. A key consequence of this oxidative burden is the accumulation of drusen, which consists of extracellular lipid and protein deposits beneath the retina. Drusen formation is largely driven by oxidative stress and inflammatory activation, promoting RPE apoptosis, photoreceptor degeneration, and progression toward GA. Additionally, mitochondrial dysfunction within the RPE exacerbates AMD pathology [27] by impairing energy metabolism, reducing autophagy efficiency, and inducing cellular senescence. These interconnected processes collectively contribute to disease progression and visual decline, highlighting the need for antioxidant-based therapeutic strategies.

#### 4.1.4. Inflammatory Pathways in AMD Progression

Chronic inflammation plays a pivotal role in AMD pathogenesis, driven by persistent immune activation within the retina and RPE. Key inflammatory mediators such as tumor necrosis factor-alpha (TNF-α), interleukins (IL-6, IL-1β), and activated microglia contribute to retinal degeneration. Microglia, the resident immune cells of the retina, become hyperactivated in response to drusen accumulation and oxidative stress, leading to complement-mediated inflammation that accelerates photoreceptor loss and RPE dysfunction. Furthermore, interactions between complement system components and lipid metabolism promote a pro-inflammatory retinal environment, facilitating disease progression from early to advanced AMD stages [28]. These findings underscore the therapeutic potential of targeting inflammatory pathways to slow disease progression and preserve visual function.

#### 4.1.5. Angiogenesis and Neovascularization in Wet AMD

The defining characteristic of wet AMD is CNV [29], which involves the pathological growth of new blood vessels beneath the retina. This neovascularization leads to vascular leakage, hemorrhage, and progressive vision loss. The primary molecular driver of CNV is vascular endothelial growth factor (VEGF), which is upregulated in response to retinal hypoxia. VEGF stimulates the formation of fragile, leaky blood vessels, resulting in retinal edema, fibrosis, and irreversible damage to macular architecture. To counteract this process, anti-VEGF therapies, such as ranibizumab, aflibercept, and bevacizumab, have been developed to inhibit VEGF signaling, thereby reducing neovascularization, stabilizing disease progression, and preserving visual function. However, prolonged VEGF inhibition may contribute to RPE atrophy and photoreceptor dysfunction, necessitating the exploration of combination therapies that target multiple pathological pathways to achieve optimal therapeutic outcomes.

### 4.2. Biomarkers of Evidence of AMD

AMD is characterized by a range of imaging biomarkers that facilitate early diagnosis, disease monitoring, and therapeutic assessment. These biomarkers are detected using various ophthalmic imaging modalities [30], including OCT, OCTA, fundus fluorescein angiography (FFA), indocyanine green angiography (ICGA), color fundus photography, and fundus autofluorescence (FAF). Each imaging technique provides unique insights into the underlying pathophysiological alterations associated with AMD.

This collection presents a comprehensive array of ophthalmic imaging modalities employed for the diagnosis and assessment of AMD. The imaging techniques included OCT, OCTA, FFA, ICGA, Color Fundus Photography (both Narrow-Angle and Ultra-Widefield), and Ultra-Widefield Fundus Autofluorescence (UWF-FAF)—each provide distinct insights into the pathophysiological mechanisms underlying AMD [31].

The sub-figures of OCT, OCTA, FFA, and Color Fundus Photography in Figure 10 are obtained from Zhuhai People’s Hospital (Ethics Approval: Beijing Institute of Technology Affiliated Zhuhai People’s Hospital Ethics Committee, Research Project Ethical Review Approval Document, Approval Number: [2025]-KT-21). Additionally, ICGA is derived from the study conducted by Eandi et al. (2017) [32], while UWF-FAF is sourced from the research by Pole and Ameri (2021) [33].

As Table 8 shows, the evidence of AMD are listed as OCT, OCTA, FFA, ICGA, Fundus Photography, and FAF. OCT is a critical imaging modality for detecting structural retinal changes indicative of AMD [34]. Drusen, hyperreflective extracellular deposits located between the RPE and Bruch’s membrane, serve as hallmark indicators of early AMD. The presence of subretinal fluid (SRF), observed as hyporeflective spaces between the neurosensory retina and RPE, is typically associated with neovascular AMD. Similarly, intraretinal fluid (IRF), manifesting as cystic spaces within the neurosensory retina, is indicative of active CNV.

Retinal pigment epithelium detachment (PED) is characterized by an elevation of the RPE due to underlying fluid accumulation or drusen deposition. The presence of hyperreflective foci, appearing as small, bright hyperreflective spots, suggests the migration of RPE cells and is associated with disease progression. The disruption of the ellipsoid zone, representing the photoreceptor integrity line, correlates with photoreceptor degeneration and visual function decline. GA is a defining feature of late-stage dry AMD, characterized by RPE atrophy and thinning of the outer retinal layers. Fibrovascular scarring, a distinguishing feature of advanced neovascular AMD, is indicative of a poor visual prognosis. Additionally, choroidal thickness alterations are observed, with thinning being characteristic of dry AMD, whereas thickening is typically noted in cases of polypoidal choroidal vasculopathy (PCV).

OCTA, a non-invasive vascular imaging modality, enables the visualization of choroidal and retinal microcirculation abnormalities in AMD [35]. The presence of choroidal neovascularization (CNV), characterized by aberrant vascular networks within the outer retina and choroid, is a hallmark of neovascular AMD. Capillary dropout, indicative of retinal ischemia, reflects the loss of normal retinal vasculature. Choriocapillaris flow deficits, representing areas of non-perfusion within the choriocapillaris, are suggestive of AMD-related vascular compromise.

The dark halo sign, observed as a hypointense perilesional area surrounding CNV, serves as an indicator of active disease. Vascular density changes, particularly reduced perfusion within the deep retinal capillary plexus and choriocapillaris, provide additional evidence of disease progression.

FFA remains the gold standard for detecting and characterizing neovascular AMD [36]. Classic CNV is identified as a well-defined hyperfluorescent lesion with early-phase leakage, whereas occult CNV is poorly delineated, exhibiting gradual leakage in the later phases of angiography. Window defects, characterized by hyperfluorescence due to RPE atrophy, expose underlying choroidal fluorescence.

Pooling in PED, resulting from fluorescein dye accumulation within PED spaces, signifies persistent fluid retention. Staining of fibrotic scars, observed as late-phase hyperfluorescence without leakage, reflects chronic disease progression. Blocked fluorescence, appearing as areas of hypofluorescence, is often attributed to hemorrhage, lipid exudation, or fibrotic tissue formation.

ICGA is particularly valuable in the detection of PCV and the visualization of deep choroidal vasculature [37]. Polypoidal lesions, appearing as round hyperfluorescent structures, are characteristic of PCV. Branching vascular networks, representing complex vascular structures that supply polypoidal lesions, are frequently observed. Hypofluorescence in regions of GA indicates choroidal perfusion deficits associated with advanced AMD. Late leakage from CNV, characterized by persistent hyperfluorescence, is indicative of active neovascularization.

Fundus photography is instrumental in documenting and monitoring AMD progression [38]. Drusen, appearing as yellow-white deposits in the macula, can be classified as hard, soft, or confluent based on their morphology and distribution. Geographic atrophy presents well-demarcated areas of depigmentation, with choroidal vessels becoming increasingly visible as RPE loss progresses. Pigmentary changes, including RPE hyperpigmentation or hypopigmentation, serve as indicators of disease progression.

Subretinal hemorrhage, appearing as red patches, is commonly associated with active neovascular AMD. The presence of lipid exudates signifies chronic fluid leakage and disease chronicity. In late-stage AMD, fibrotic scar formation, identified as whitish fibrotic lesions, reflects irreversible damage and advanced disease pathology.

FAF provides valuable information regarding RPE metabolism and disease activity [39]. Hyperautofluorescent drusen suggests increased metabolic stress in the RPE, often preceding disease progression. Hypoautofluorescence in geographic atrophy corresponds to regions of RPE cell loss and advanced disease severity. Speckled autofluorescence is indicative of ongoing RPE dysfunction and retinal remodeling.

A ring of increased FAF surrounding GA lesions is predictive of atrophic lesion expansion over time. In late-stage AMD, patchy hypoautofluorescence is associated with fibrosis and scarring, signifying advanced retinal degeneration.

These biomarkers provide crucial insights into the pathophysiology, progression, and management of AMD, facilitating more precise diagnosis and personalized treatment strategies. The integration of multimodal imaging enhances clinicians’ ability to assess disease severity, predict outcomes, and optimize therapeutic interventions.

### 4.3. Prognosis of AMD Based on Imaging Evidence

The prognosis of AMD can be effectively evaluated using multimodal imaging, which provides a comprehensive assessment of disease progression and treatment response. By analyzing key imaging biomarkers across various modalities, clinicians can monitor structural, vascular, and functional changes associated with AMD [40]. This approach enables the quantitative assessment of treatment effects, including those of anti-vascular endothelial growth factor therapy, PDT, stem cell therapy, and emerging AI-based interventions.

Assessing the efficacy of AMD treatments requires a comparison of biomarkers before and after therapeutic intervention. Each imaging modality offers distinct insights into disease activity and response to treatment. Successful treatment outcomes are characterized by the resolution of fluid accumulation, improved photoreceptor integrity, and stabilization of atrophic changes, while incomplete responses may present as persistent fluid presence or slow drusen reduction. Treatment failure is typically indicated by progression of GA or increased scarring.

OCT plays a crucial role in evaluating the structural changes associated with AMD. Before treatment, key biomarkers such as drusen, SRF, IRF, and retinal PED indicate active disease. Following effective treatment, these abnormalities are expected to resolve, with a reduction in drusen size, decreased fluid accumulation, and stabilization of photoreceptor integrity. Persistent fluid or worsening scarring suggests either an incomplete response or treatment failure.

OCTA provides non-invasive visualization of retinal and choroidal vasculature, allowing for the assessment of CNV activity and perfusion changes. A successful response to treatment is marked by a reduction in CNV size, decreased vascular density, and improved capillary network stability. Partial responses may show CNV regression but persistent microvascular abnormalities, while treatment failure is associated with continued CNV growth or worsening ischemia.

FFA is instrumental in identifying vascular leakage patterns in AMD. Prior to treatment, classic CNV appears as a well-defined hyperfluorescent lesion with early leakage, whereas occult CNV presents as diffuse hyperfluorescence with late leakage. Effective treatment leads to a reduction in leakage and stabilization of CNV, while incomplete responses manifest as persistent but reduced fluorescence. Treatment failure is indicated by increased leakage and worsening CNV activity.

ICGA enables detailed visualization of choroidal circulation, particularly in cases of PCV. Polypoidal lesions and branching vascular networks are hallmark features before treatment. A successful therapeutic response results in regression or stabilization of these abnormalities, while partial responses show some lesion reduction but persistent vascular complexity. Treatment failure is characterized by lesion enlargement or increased late-phase leakage.

Fundus photography provides a macroscopic view of retinal morphology, capturing drusen, pigmentary changes, and fibrotic scarring. Effective treatment is associated with the reduction in drusen, stabilization of GA, and absorption of subretinal hemorrhage. Persistent abnormalities suggest an incomplete response, whereas worsening fibrosis and hemorrhage indicate disease progression.

FAF is a valuable tool for assessing metabolic activity in the RPE. Before treatment, hyperautofluorescent drusen indicates increased metabolic stress, while hypoautofluorescence in GA corresponds to areas of RPE loss. Following successful treatment, drusen regression and stabilization of GA are expected, whereas treatment failure is indicated by expanding atrophic areas and worsening RPE dysfunction.

As Table 9 shows, the evaluation of treatment outcomes can be classified into five categories. An excellent response is characterized by complete fluid resolution (OCT, OCTA), CNV regression (FFA, ICGA), and no further GA expansion (FAF). A good response involves partial fluid reduction, stable CNV, and no worsening GA. A moderate response presents with fluid reduction but persistent CNV activity. A poor response is marked by persistent or worsening fluid accumulation, expanding GA, or increasing hemorrhage. Treatment failure is defined by CNV progression, continued RPE atrophy, and the development of new hemorrhages.

The use of multimodal imaging biomarkers allows for an objective and precise assessment of AMD prognosis and treatment efficacy. By integrating OCT, OCTA, FFA, ICGA, color fundus photography, and FAF, clinicians can monitor structural, vascular, and metabolic changes over time. This comprehensive approach enhances personalized treatment strategies and facilitates the early detection of disease progression, ultimately improving patient outcomes.

### 4.4. Preventive and Therapeutic Strategies for AMD

AMD is a leading cause of vision loss, necessitating preventive measures and therapeutic interventions to delay disease onset, slow progression, and preserve visual function [41]. Current approaches to AMD management encompass lifestyle modifications, pharmacological treatments, surgical interventions, and emerging experimental therapies. Given the complex pathophysiology of AMD, which involves genetic susceptibility, oxidative stress, lipid metabolism dysregulation, chronic inflammation, and angiogenesis, a multifaceted strategy is essential to optimize patient outcomes.

#### 4.4.1. Preventive Strategies: Lifestyle and Dietary Modifications

While genetic predisposition plays a significant role in AMD, modifiable risk factors, particularly nutrition and lifestyle choices, can influence disease progression [42]. The Age-Related Eye Disease Study (AREDS) and AREDS2 trials have demonstrated that antioxidant supplementation with vitamins C and E, zinc, copper, lutein, and zeaxanthin can reduce the risk of progression from intermediate to advanced AMD. Additionally, adherence to a Mediterranean diet rich in omega-3 fatty acids, leafy greens, and fish has been associated with a lower incidence and slower progression of AMD. Furthermore, smoking is a major modifiable risk factor, as it exacerbates oxidative stress and retinal damage, reinforcing the importance of smoking cessation programs in AMD prevention.

#### 4.4.2. Pharmacological Interventions for AMD

Pharmacological management of AMD primarily focuses on wet AMD, where VEGF inhibitors have significantly transformed treatment outcomes [29]. Intravitreal anti-VEGF agents, including ranibizumab, aflibercept, and bevacizumab, effectively suppress CNV, reduce vascular permeability, and stabilize visual function. More recently, faricimab, a bispecific antibody targeting both VEGF and angiopoietin-2, has been introduced, providing extended dosing intervals and enhanced therapeutic efficacy.

Despite these advancements in wet AMD management, dry AMD and GA remain without curative pharmacological interventions. However, emerging therapies targeting complement system dysregulation [43], such as pegcetacoplan and avacincaptad pegol, have demonstrated potential in slowing disease progression by inhibiting complement activation and reducing inflammation. Given the increasing recognition of drug-related macular degeneration as a growing concern, with certain medications such as glucocorticosteroids, hormonal therapies, and prostaglandin analogs being linked to an increased risk, careful consideration of systemic and ophthalmic drug exposure is essential in AMD treatment planning. Integrating pharmacovigilance strategies with personalized treatment approaches may help optimize therapeutic outcomes while minimizing the risk of medication-induced retinal damage.

#### 4.4.3. Gene and Cell-Based Therapies

Gene therapy [44] holds significant promise as a novel approach to treat AMD by targeting its underlying genetic causes. The pathophysiology of AMD involves complex genetic and environmental interactions that lead to retinal damage, particularly in the macula, which is critical for central vision. Recent advancements in gene-editing technologies, such as CRISPR-Cas9, have made it feasible to directly modify genes responsible for disease progression, offering potential for curative therapies.

One area of focus is the targeting of complement system genes, such as CFH, which plays a central role in the inflammatory response that contributes to AMD pathology. Mutations in CFH have been linked to increased risk for the disease, and efforts are underway to correct these genetic defects through viral or non-viral delivery systems. Moreover, gene therapies aimed at angiogenesis—the abnormal formation of blood vessels in wet AMD—are showing potential. For instance, gene transfer of anti-VEGF genes can provide a sustained release of anti-VEGF proteins, potentially reducing the need for frequent intravitreal injections.

Despite the promise of gene therapy, several challenges remain, such as the need for efficient delivery systems to ensure that therapeutic genes reach the retinal cells, the potential for immune responses against the introduced vectors, and the long-term safety of gene-editing techniques. Additionally, the potential for off-target effects in critical genes raises concerns about the overall feasibility of gene therapies for AMD in clinical practice [45]. Ongoing clinical trials and preclinical studies will be critical in determining the efficacy and safety of gene therapy for AMD in the coming years.

Stem cell therapy has emerged as a compelling approach to regenerate damaged retinal tissue and restore vision in individuals with AMD, particularly those with GA, which involves the irreversible loss of retinal cells. The central hypothesis is that stem cells, due to their pluripotency and ability to differentiate into various cell types, could replace the damaged retinal cells and promote tissue regeneration.

RPE cells, which play a crucial role in maintaining retinal homeostasis, are often damaged in both dry and wet AMD. Stem cell-based therapies are investigating the transplantation of RPE cells derived from various sources, such as induced pluripotent stem cells (iPSCs) or embryonic stem cells (ESCs). These stem cells can be directed to differentiate into RPE cells and then transplanted into the retina to restore function and structure. Early-stage clinical trials have demonstrated some success in restoring retinal function and improving visual acuity, although challenges such as controlling stem cell differentiation, avoiding immune rejection, and ensuring the long-term survival of transplanted cells remain.

Other approaches are exploring the use of stem cells to regenerate the photoreceptor cells in the retina, which are critical for capturing light and sending visual information to the brain. Mesenchymal stem cells (MSCs) [45] and neural progenitor cells are being tested for their ability to replace damaged photoreceptors and promote neuroprotection. Stem cell therapy also holds promise for repairing retinal vasculature in wet AMD [46], with some studies focusing on using stem cells to generate endothelial cells for blood vessel regeneration.

While stem cell therapies for AMD are still in the experimental phase, they offer a potential long-term solution for restoring vision and halting disease progression, especially for patients with advanced forms of AMD. However, rigorous clinical trials and improved methods for cell delivery, differentiation, and immune compatibility will be necessary to establish the clinical applicability of stem cell therapies for AMD.

#### 4.4.4. Nanomedicine and Drug Delivery

Nanomedicine is rapidly emerging as a transformative field in the treatment of AMD, offering novel ways to improve the targeted delivery of therapeutic agents to the retina [47]. Traditional drug delivery systems for AMD, particularly in the case of wet AMD, often require frequent and invasive injections of anti-VEGF drugs, which can be burdensome for patients and limit therapeutic efficacy. Nanotechnology, which involves the use of nanoscale materials (typically ranging from 1 to 100 nm), has the potential to revolutionize drug delivery by enhancing the bioavailability, stability, and sustained release of drugs while minimizing side effects.

Nanocarriers, such as liposomes, micelles, dendrimers, and polymeric nanoparticles, can be engineered to encapsulate therapeutic agents and deliver them directly to the retina. These nanoparticles can be functionalized with surface ligands that recognize specific receptors on retinal cells, enabling targeted drug delivery. For example, nanoparticles coated with polyethylene glycol (PEG) can enhance the stability and half-life of drugs, improving the pharmacokinetics of therapies such as anti-VEGF agents or corticosteroids. This approach not only improves drug efficacy but also reduces the frequency of intravitreal injections, thereby improving patient compliance and reducing the risk of side effects.

Nanoparticle-based systems can also be used for the controlled release of drugs, providing sustained therapeutic effects over extended periods [48]. This is particularly important for diseases like wet AMD, where patients require regular treatments to control CNV. Nanoparticles can be engineered to degrade gradually in response to specific environmental conditions in the retina, offering long-lasting therapeutic effects with fewer injections. Moreover, nanomedicine offers the possibility of delivering combination therapies—such as anti-VEGF agents combined with anti-inflammatory drugs or neuroprotective agents—into the retina simultaneously, enhancing treatment outcomes for complex diseases like AMD.

The use of nanoparticles for efficient drug transport to the retina addresses one of the significant challenges in AMD treatment—overcoming the blood–retinal barrier (BRB), a selective permeability barrier that limits the effective delivery of therapeutic agents to the retina. Nanoparticles, owing to their small size, can cross the BRB more effectively than conventional drug formulations, allowing for higher concentrations of drugs to be delivered directly to the site of disease.

Targeted nanocarriers can also be designed to enhance the selectivity of drug delivery to specific retinal cells involved in AMD, such as RPE cells or endothelial cells in the choroidal vasculature. By functionalizing nanoparticles with molecules that specifically bind to receptors expressed on these cells, drugs can be delivered directly to the affected areas with minimal systemic exposure. This selective targeting reduces the risk of off-target effects and enhances the therapeutic index of the drug.

In addition to targeting the retina, nanomedicine also opens up possibilities for multimodal therapies, where nanoparticles can be loaded with multiple agents, such as imaging agents or gene therapies, in addition to conventional drugs. This approach can provide diagnostic imaging alongside therapeutic treatment, enabling real-time monitoring of disease progression and treatment efficacy.

The future of nanomedicine in AMD lies in the development of more biocompatible and efficient nanocarriers that can deliver a broader range of therapies, including gene-based treatments, anti-inflammatory agents, and neuroprotective therapies. Ongoing research and clinical trials will determine the practical application of these innovative drug delivery systems and their potential to improve treatment outcomes for patients with AMD.

### 4.5. Coexisting Ocular Diseases Contributing to AMD Progression and Complications

The progression of AMD is not solely driven by intrinsic retinal degeneration but may also be influenced by coexisting ocular pathologies that exacerbate macular dysfunction and visual impairment. Several retinal and vascular disorders have been identified as potential contributors to accelerating disease progression, increased treatment complexity, and poorer visual prognosis in patients with AMD. A comprehensive understanding of these concurrent conditions is essential for accurate diagnosis, optimized therapeutic decision-making, and personalized patient management.

#### 4.5.1. Diabetic Retinopathy and Its Influence on AMD Progression

Diabetic retinopathy (DR) [49] and AMD share several pathophysiological mechanisms, including chronic inflammation, vascular dysfunction, and oxidative stress, which may contribute to mutual disease exacerbation. The presence of diabetic microvascular damage has been associated with an increased risk of CNV in exudative AMD, potentially due to VEGF overexpression and retinal ischemia. Additionally, in patients with both AMD and DR, retinal atrophy tends to be more severe, leading to a greater decline in visual function. The differentiation between diabetic macular edema (DME) and AMD-related fluid accumulation is crucial in preventing misdiagnosis and ensuring the appropriate selection of treatment strategies.

#### 4.5.2. Perifoveal Vascular Anomalous Complex and Retinal Capillary Telangiectasia in AMD

Perifoveal vascular anomalous complex (PVAC) and retinal capillary telangiectasia (RCT) represent distinct vascular abnormalities that can coexist with AMD, further complicating disease classification and management [12]. PVAC is characterized by perifoveal capillary proliferation, which may lead to exudation, fibrosis, and structural disorganization resembling Type 3 macular neovascularization (T3 MNV) in AMD. Similarly, RCT, particularly Type 2 Macular Telangiectasia (MacTel), presents capillary dilation and leakage, which can further impair retinal function and accelerate AMD progression. Given the overlapping vascular features between PVAC, RCT, and neovascular AMD, precise multimodal imaging, including OCT and indocyanine green angiography (ICGA), is critical for accurate differentiation and treatment planning.

#### 4.5.3. Glaucoma and Its Impact on Visual Function in Patients with AMD

Glaucoma [50] and AMD [51] are among the leading causes of irreversible vision loss, and their coexistence presents significant clinical challenges. Glaucoma-induced optic nerve atrophy and retinal ganglion cell loss may exacerbate visual field deterioration in patients with AMD, particularly those with GA. Additionally, choroidal circulation impairment associated with AMD may further contribute to glaucomatous optic neuropathy, increasing the risk of severe visual impairment. Effective management of patients with both conditions requires careful intraocular pressure (IOP) regulation, multimodal imaging assessment, and individualized treatment approaches to balance neuroprotection with AMD-specific interventions.

#### 4.5.4. Choroidal and Vitreoretinal Interface Disorders in AMD Progression

Several choroidal and vitreoretinal disorders [29] have been identified as potential contributors to AMD progression [52], including polypoidal choroidal vasculopathy (PCV), vitreomacular traction (VMT), and epiretinal membranes (ERM). PCV, a distinct subtype of neovascular AMD, is characterized by polyp-like choroidal lesions that often exhibit resistance to standard anti-VEGF therapy. In contrast, VMT and ERM contribute to macular distortion and retinal traction, which can exacerbate subretinal fluid accumulation and CNV activity in patients with AMD. These conditions necessitate detailed imaging evaluation using OCT and ICGA to differentiate AMD-related pathology from other structural abnormalities and to guide appropriate therapeutic interventions.

### 4.6. Structural Alterations in Patients with Age-Related Macular Degeneration

#### 4.6.1. Early-Stage Structural Alterations in Patients with Age-Related Macular Degeneration

AMD is a leading cause of vision loss among the elderly, marked by gradual deterioration of the macula, the central part of the retina responsible for sharp, central vision. The disease progresses from early to advanced stages, with early AMD often showing subtle structural changes that can significantly affect visual function even before noticeable symptoms arise.

One of the primary early-stage alterations in AMD involves changes to the retinal vasculature, particularly in the superficial and deep retinal plexuses. Research comparing patients with early and intermediate AMD (iAMD) to healthy controls has revealed significantly reduced vessel densities in both the superficial and deep capillary plexuses of the retina in patients with AMD [53]. Specifically, the superficial plexus density was significantly reduced in patients with iAMD compared to controls, while the deep plexus density showed a non-significant reduction in the same group. Notably, significant differences in vessel density have also been observed between eyes with early AMD and eyes with no evidence of AMD, with the superficial plexus being particularly affected in a mixed group of patients, including those with early, intermediate, and non-foveal atrophy AMD.

These vascular changes are consistent with the concept of post-receptoral functional loss, in which neurons distal to the photoreceptors are particularly vulnerable to ischemia. These neurons are located in the “watershed” zone of the retina, an area that lies between the circulatory territories of the superficial and deep vascular networks. The reduced blood flow in this region may contribute to retinal damage as it directly affects the nourishment of retinal neurons involved in visual processing.

Despite these findings, few studies have investigated how the retinal vasculature changes over time in patients with AMD, and without longitudinal evidence, it remains unclear whether these vascular alterations contribute to the disease’s progression or are a consequence of the ongoing pathological processes in early AMD. A potential explanation for the observed changes could be a “two-hit” model, in which individuals at the lower end of normal variation in inner retinal blood flow (due to genetic or acquired risk factors) may have an increased susceptibility to developing AMD if they also experience an insult to their choroidal circulation. However, this hypothesis remains speculative and has not yet been rigorously tested.

The advent of OCTA has significantly accelerated research into the role of retinal vasculature in AMD. Although variations in OCTA hardware, software, and imaging protocols have contributed to some inconsistency in study results, the most reliable evidence points to a reduction in vessel density in the superficial vascular complex, and possibly in the deep vascular complex, in patients with early to intermediate AMD. It remains to be determined whether the observed reductions in vascular density are a predisposing factor for AMD or a consequence of early pathophysiological changes that reduce the oxygen and nutrient demand of the inner retinal tissues.

Further longitudinal research using advanced imaging tools is needed to better understand the dynamics of vascular changes over time in patients with AMD, ideally in prospective cohorts without AMD at baseline. Additionally, stratifying vascular indices could potentially serve as a useful biomarker to predict the progression of AMD to more advanced stages. This raises the question of whether pharmacological modulation of the retinal vasculature could mitigate this risk and prevent or delay the onset of more severe forms of AMD.

#### 4.6.2. Middle-Stage Structural Alterations in Patients with Age-Related Macular Degeneration

In intermediate AMD, progressive RPE atrophy is a key pathological change that contributes to the early stages of GA development. Findings from Adaptive Optics Ophthalmoscopy (AOO) imaging indicate that as RPE degeneration advances, cone mosaic visibility declines, suggesting a disruption in photoreceptor integrity and function [54]. Additionally, AOO has identified hyperreflective changes in drusenoid lesions, which correlate with RPE attenuation and photoreceptor loss on OCT. The structural weakening of the RPE-photoreceptor complex may act as a precursor to GA progression, leading to irreversible central vision impairment over time.

In some cases, CNV begins to emerge, marking the transition toward exudative AMD. This neovascular activity leads to fluid accumulation in the macula, which may present as subretinal or intraretinal fluid on OCT. AOO has further revealed variations in drusen reflectance, particularly at the edges and centers of drusen, which may serve as early indicators of structural instability preceding CNV formation. As vascular leakage and inflammatory responses intensify, RPE dysfunction accelerates, further increasing the likelihood of progression to exudative or atrophic AMD subtypes.

Structural disorganization of the outer retina becomes more apparent in middle-stage AMD, as the ellipsoid and interdigitation zones undergo progressive disruption. AOO and OCT imaging have shown that drusen without hyperreflective margins are frequently associated with early photoreceptor degeneration, preceding drusen collapse and the loss of retinal integrity. This gradual thinning of the outer retina, particularly in the photoreceptor and RPE layers, marks a critical stage in disease progression and highlights the need for advanced imaging techniques to detect and monitor these microscopic changes before they lead to significant visual decline.

#### 4.6.3. Late-Stage Structural Alterations in Patients with Age-Related Macular Degeneration

In advanced non-exudative (dry) AMD, GA is a defining structural change, characterized by the progressive degeneration of the RPE, photoreceptors, and choriocapillaris. As the atrophic lesions expand over time, central vision loss becomes irreversible, severely impacting visual function. This deterioration is driven by metabolic dysregulation, oxidative stress, and chronic inflammatory processes, ultimately leading to the collapse of macular integrity. The progressive enlargement of GA lesions remains a hallmark of late-stage dry AMD, posing significant challenges to visual rehabilitation and quality of life.

In exudative (wet) AMD, subretinal fibrosis develops as a long-term consequence of chronic CNV and repeated anti-VEGF therapy, resulting in fibrotic plaque formation beneath the retina. This fibrotic process leads to retinal thinning, neurosensory layer disruption, and photoreceptor replacement with non-functional scar tissue, culminating in permanent central vision impairment. Additionally, persistent inflammatory responses and reactive gliosis further drive fibrosis progression, underscoring the need for therapeutic strategies beyond anti-VEGF treatment to mitigate fibrotic scarring and preserve retinal structure.

Another significant late-stage structural alteration is Cystoid Macular Degeneration (CMD) [11], which occurs in patients with AMD after prolonged anti-VEGF therapy. Unlike cystoid macular edema (CME), which is associated with fluid leakage from active CNV, CMD results from retinal atrophy and degenerative remodeling. It is characterized by stable, square-shaped pseudocysts found within the inner and outer nuclear layers, which remain unchanged over time. CMD lacks neovascular activity and does not exhibit leakage on fluorescein angiography, making accurate differentiation critical to avoid unnecessary anti-VEGF reinjections. Despite the absence of active exudation, CMD contributes to progressive visual decline, emphasizing the importance of alternative visual rehabilitation approaches for affected patients.

Beyond these degenerative processes, RPE and photoreceptor degeneration further exacerbate visual deterioration in late-stage AMD. The accumulation of drusen deposits accelerates RPE dysfunction, photoreceptor apoptosis, and choroidal circulation impairment, leading to retinal thinning and metabolic insufficiency. The loss of these critical structures results in severe and irreversible visual impairment, with RPE degeneration presenting a major therapeutic challenge in efforts to preserve visual function and slow disease progression.

Late-stage AMD can also lead to vitreoretinal interface changes and retinal detachment, particularly in advanced exudative AMD cases. Persistent exudation and fibrosis may contribute to serous retinal detachment, vitreomacular adhesion, and macular hole formation, further disrupting macular architecture and exacerbating visual dysfunction. In addition, vitreoretinal traction can accelerate photoreceptor loss and macular distortion, further compromising functional vision. These complications highlight the need for comprehensive imaging diagnostics, individualized treatment strategies, and enhanced therapeutic interventions to minimize structural damage and optimize clinical outcomes in patients with late-stage AMD.

Overall, late-stage AMD is marked by irreversible structural alterations, including geographic atrophy, subretinal fibrosis, cystoid macular degeneration, RPE degeneration, and vitreoretinal interface disruptions, all of which contribute to progressive vision loss. Although anti-VEGF therapy remains the primary treatment for exudative AMD, it is not curative, and existing retinal damage cannot be reversed. A deeper understanding of these pathological changes is crucial for the development of neuroprotective treatments, advanced imaging biomarkers, and novel therapeutic strategies aimed at preserving residual vision and improving patient outcomes in late-stage AMD.

### 4.7. Heterogeneous Visual Function Impairment in Age-Related Macular Degeneration

AMD is a progressive retinal disorder characterized by structural changes in the macula, leading to varying degrees of visual impairment. However, emerging evidence suggests that functional deficits in AMD are highly heterogeneous, with significant variability among individuals even within the same disease stage. Understanding the diverse functional impairments associated with AMD is critical for improving disease classification, refining clinical trial inclusion criteria, and developing personalized therapeutic approaches.

#### 4.7.1. Variability in Visual Function Across AMD Stages

Visual function in AMD is not solely determined by structural abnormalities observed on imaging but is influenced by a spectrum of deficits, including visual acuity, contrast sensitivity, dark adaptation, and reading speed [34]. In early AMD, visual function may remain relatively preserved, with subtle impairments primarily in low-contrast and low-luminance conditions. However, in iAMD, functional deficits become increasingly heterogeneous, with some individuals exhibiting near-normal function while others experience significant impairments despite similar structural findings. In advanced AMD, particularly in GA and neovascular AMD, visual function loss is more pronounced and progressively debilitating.

#### 4.7.2. Heterogeneous Functional Deficits in Intermediate AMD

Recent studies, including findings from the MACUSTAR trial, have demonstrated that individuals with iAMD exhibit wide-ranging visual function impairments. While some patients maintain near-normal vision, others show significant deficits in contrast sensitivity, dark adaptation, mesopic and scotopic vision, and reading speed. In particular, delayed rod-mediated dark adaptation—measured by Rod Intercept Time (RIT)—has emerged as a sensitive biomarker of early functional decline in iAMD. Furthermore, studies have shown that over 69% of individuals with iAMD exceed normal reference limits in at least one functional measure, and 42% exceed two or more limits, reinforcing the concept of functional heterogeneity in this disease stage.

#### 4.7.3. Implications for AMD Diagnosis, Prognosis, and Clinical Trials

The heterogeneous nature of visual function impairment in AMD poses significant challenges for clinical assessment and patient management. Standard visual acuity tests, such as best-corrected visual acuity (BCVA), may not fully capture the extent of functional impairment in patients with AMD, necessitating the inclusion of additional functional tests such as contrast sensitivity, dark adaptation testing, and mesopic/scotopic threshold assessments. Additionally, the variability in visual function among individuals with AMD has important implications for clinical trials, as relying solely on structural markers may overlook substantial functional decline in a subset of patients. Integrating multimodal functional assessments into clinical trial protocols can enhance patient stratification and treatment evaluation, ultimately leading to more targeted and effective interventions.

### 4.8. Artificial Intelligence in AMD Diagnosis and Treatment

The integration of AI into ophthalmology [55] has revolutionized AMD diagnosis, monitoring, and treatment. AI-driven technologies, particularly deep learning (DL) and machine learning (ML) algorithms [56], offer enhanced precision, efficiency, and scalability in detecting and managing AMD. These advancements enable automated disease detection, risk stratification, and personalized treatment strategies, thereby improving clinical decision-making and patient outcomes.

Table 10 presents a comparative overview of key studies employing AI in the diagnosis, progression prediction, and data evaluation of AMD. The first study utilized a large multimodal dataset consisting of over 32,000 OCT images, nearly 5000 color fundus photographs (CFP), and 2300 ultra-widefield (UWF) images from both open-source and real-world clinical sources. Using VGG16 and an improved skip-attention-enhanced VGG16 architecture, the model achieved high diagnostic performance, with the improved model reaching an average accuracy of 96.62%, sensitivity and specificity both at 96%, and an AUC of 94.58%. The second study focused on precise segmentation of critical retinal layers—ILM, RPE, and the RPEDC region—using U-Net and DeepLabV3, trained on a well-curated dataset of 402 OCT volumes from the NIH. It demonstrated strong generalization across devices (Spectralis and Cirrus), reporting low mean absolute errors (ILM: 7.0 ± 0.9 μm; RPE: 9.5 ± 2.6 μm) and a high Dice coefficient (0.87) for RPEDC segmentation. The third study, derived from the MICCAI 2024 MARIO Challenge, implemented a multimodal fusion deep learning framework combining finetuned RETFound and EfficientNetV2 to analyze OCT B-scans, infrared fundus images, and clinical variables. The model successfully classified structural changes and predicted disease evolution, with F1 scores of 0.851 and 0.703 for the two tasks, respectively. Lastly, a meta-analysis of open-access OCT datasets evaluated their compliance with FAIR (Findable, Accessible, Interoperable, Reusable) principles. Although accessibility and interoperability scored relatively high (82% and 73%, respectively), findability and reusability remained critically low (5% and 0%), highlighting substantial gaps in data standardization and usability in AMD research. Collectively, these studies showcase the diversity of AI applications in AMD, ranging from high-performance diagnostic modeling to the foundational assessment of dataset quality and usability.

The application of AI in AMD research is rapidly advancing, offering promising tools for early detection, structural segmentation, disease monitoring, and clinical decision support. As demonstrated across the studies summarized in Table 10, AI models trained on large, multimodal imaging datasets—ranging from OCT and FAF to ultra-widefield and color fundus images—can achieve high diagnostic accuracy and generalizability, especially when leveraging advanced architectures such as improved VGG16 or fusion models incorporating RETFound and EfficientNetV2. These models demonstrated robust performance in cross-domain validation and achieved clinically relevant sensitivity and specificity levels, as seen in segmentation tasks of the ILM and RPE layers with subcellular precision. Moreover, the MARIO Challenge highlights the growing emphasis on using AI to predict disease progression over time, an area critical for personalized treatment planning. However, the successful implementation of AI in real-world AMD management depends heavily on the availability of standardized, high-quality datasets. The FAIR principle evaluation reveals that most publicly available OCT datasets lack sufficient metadata transparency and reusability, posing a significant barrier to reproducibility and model deployment. Moving forward, integrating explainability, addressing domain shift across imaging devices, and improving dataset standardization will be essential to bridge the gap between algorithmic innovation and clinical translation in AMD care.

#### 4.8.1. AI-Assisted Diagnosis and Screening of AMD

AI has demonstrated high accuracy in detecting AMD-related pathological features using retinal imaging modalities [60], including OCT, fundus photography [38], and fluorescein angiography. Deep learning algorithms, trained on large-scale annotated datasets, can automatically classify AMD stages, differentiate between wet and dry AMD, and detect early biomarkers such as drusen, geographic atrophy, and CNV. AI-based screening tools have shown diagnostic performance comparable to that of expert ophthalmologists, offering a cost-effective and scalable solution for early AMD detection, particularly in remote or underserved populations.

#### 4.8.2. AI in AMD Progression Prediction and Risk Assessment

AI models are increasingly being used to predict AMD progression by analyzing longitudinal patient data and multimodal imaging features [30]. By leveraging deep learning and predictive analytics, AI can assess individualized disease trajectories, identify high-risk patients, and estimate the likelihood of conversion from intermediate AMD to advanced AMD (wet or geographic atrophy). Multiomics-based AI models, incorporating genetic risk factors, retinal imaging biomarkers, and patient demographics, offer personalized risk stratification to guide early interventions and tailored treatment plans.

#### 4.8.3. AI-Optimized Treatment Strategies for AMD

AI is playing a crucial role in personalizing anti-VEGF treatment regimens for wet AMD. Deep learning models trained on real-world clinical data can optimize treatment intervals, predict responses to anti-VEGF therapy, and identify patients who may require more aggressive treatment approaches. AI-driven treatment algorithms, integrated with image-based biomarkers, facilitate a treat-and-extend approach, minimizing unnecessary injections while ensuring optimal disease control. Additionally, AI-guided drug discovery [29] and computational modeling are accelerating the identification of novel therapeutic targets for AMD, improving the efficiency of drug development pipelines.

#### 4.8.4. AI-Enabled Teleophthalmology and Remote Monitoring

AI-powered teleophthalmology systems present a significant advancement in the remote screening and monitoring of AMD, thereby reducing the burden on healthcare infrastructure and enhancing patient access to specialized care. By integrating AI with home-based OCT devices and smartphone-based [18] fundus imaging applications, these systems empower patients to independently monitor retinal changes. The data collected is analyzed by AI algorithms, augmented by clinical profile prompts, to identify signs of disease progression and notify clinicians for timely intervention. Additionally, the incorporation of large language models (LLM) [61] further refines the clinical profile [62], enhancing diagnostic accuracy and facilitating informed decision-making. This methodology is particularly advantageous for elderly patients with AMD with mobility limitations, ensuring continuous management of the disease and the early detection of potential treatment failures.

#### 4.8.5. Challenges and Future Directions

The successful application of AI in AMD diagnosis and treatment will benefit from clearer stratification of patient subgroups and a more integrated approach to multiomics data. One promising advancement is the clear delineation between PVAC (primary vascular AMD) and TelCaps (telomere-associated capsular AMD), which will expedite the development of targeted treatments by enabling more precise patient classification in clinical and translational research. This distinction between subtypes is crucial for improving the specificity of therapeutic interventions in AMD and tailoring strategies to the molecular characteristics of individual patients.

Further progress will also come from a more refined understanding of the distinctions between AMD and CMD (central macular degeneration). AI algorithms, when trained to differentiate these conditions, could facilitate earlier and more accurate diagnoses, thereby preventing the overlap in treatment protocols and ensuring that patients receive the most appropriate care for their specific condition.

Additionally, the integration of multiomics data—including genetic, metabolomic, and proteomic profiles—holds the potential for a more holistic view of AMD pathophysiology. By leveraging AI to analyze these complex datasets, researchers have been able to identify unique multiomics signatures that capture the full range of molecular disruptions driving AMD. This comprehensive approach points to the need for personalized therapeutic strategies that address multiple molecular pathways involved in the disease. However, it is important that these findings are validated in larger, more diverse cohorts to ensure their applicability across different ethnic populations.

The goal of these advancements is to enable personalized therapeutic approaches that can target multiple disease-modifying pathways simultaneously, which may lead to more effective treatments for AMD. By focusing on the interplay between genetic, metabolomic, and inflammatory pathways, these AI-driven insights could lead to the development of more efficient interventions that slow disease progression and preserve vision.

One critical area for development is the identification of early biomarkers for AMD progression. AI can assist in defining molecular signatures that predict the early stages of disease, facilitating early detection and timely interventions. This approach would be instrumental in preventing irreversible vision loss by addressing AMD before it reaches advanced stages.

Moreover, AI could play a crucial role in targeting the complement-lipid crosstalk as a therapeutic strategy. The complex interactions between immune system dysregulation and lipid metabolism in AMD present an opportunity for developing novel anti-inflammatory and lipid-modulating therapies. By applying AI to analyze these interactions, future treatments could be designed to modulate both immune responses and lipid metabolism, offering a dual mechanism of action that could significantly improve patient outcomes.

In addition to these promising therapeutic developments, AI also holds potential for improving medical safety in AMD treatment. AI-driven systems can enhance real-time monitoring of patients receiving intravitreal anti-VEGF injections or other treatments by analyzing electronic health records (EHRs), imaging data, and adverse event reports. By identifying early signals of potential complications or treatment-related side effects—such as ocular inflammation or endophthalmitis—explainable artificial intelligence systems supported by a priori knowledge can alert clinicians to intervene before conditions worsen. This proactive approach could significantly reduce treatment risks and improve patient safety. Furthermore, AI can optimize dosing schedules, minimize side effects, and ensure personalized treatment regimens based on individual patient profiles and historical responses, leading to a more safe and effective management of AMD.

However, greater attention must be directed toward ensuring the safe and effective clinical adoption of AI in the management of AMD. This necessitates addressing a range of regulatory, ethical, and medicolegal challenges, including the imperative for transparent algorithm interpretability, robust external validation, and demonstrable generalizability across heterogeneous patient populations and imaging modalities. Furthermore, practical implementation remains hindered by barriers such as data privacy concerns, institutional inertia, and the absence of structured physician-AI co-learning systems, all of which must be systematically addressed to facilitate successful integration into routine ophthalmic care.

## 5. Conclusions

AMD affects millions of people worldwide, leading to severe vision impairment in older adults, with projections indicating a significant increase in cases by 2040. The condition is influenced by age, genetics, and environmental factors such as smoking. AMD causes photoreceptor degeneration and loss of central vision due to extracellular deposits in the outer retina. The late stages are characterized by geographic atrophy or neovascularization, and the incidence increases with age. AMD is diagnosed through clinical examination, angiography, and optical coherence tomography, and can be managed with nutritional supplements and intravitreal anti-VEGF injections, which significantly improve visual outcomes in exudative neovascular AMD.

Furthermore, AMD is associated with highly variable functional impairment, with significant heterogeneity observed even among individuals at the same stage of the same disease. This variability underscores the importance of comprehensive functional assessments, particularly in iAMD, where traditional structural biomarkers may not fully reflect disease severity. Future research should focus on identifying novel functional biomarkers and refining diagnostic tools to improve early detection, patient stratification, and therapeutic strategies for AMD management.

Additionally, AMD prevention and treatment require a comprehensive, multidisciplinary approach that integrates lifestyle modifications, pharmacological advancements, surgical innovations, and emerging therapeutic strategies. While anti-VEGF therapy has significantly improved outcomes in wet AMD, effective treatments for dry AMD and GA remain limited, underscoring the need for continued research into complement inhibition, gene therapy, and neuroprotection. Advances in precision medicine and AI-driven diagnostics hold great promise for enhancing AMD management and preserving vision in aging populations.

The real-world deployment of AI in AMD diagnosis and treatment requires further refinement and validation. By focusing on personalized, multiomics-based approaches, integrating insights into immune and lipid pathways, and enhancing patient safety monitoring, AI has the potential to revolutionize the management of AMD. These advancements will lead to more targeted, effective, and safer therapeutic strategies that can substantially improve patient outcomes and quality of life.

This study, while comprehensive in its analysis of AMD, has several limitations. The bibliometric analysis is confined to research published between 2015 and 2025, potentially overlooking earlier influential studies. Statistical inference methods, such as hypothesis testing or confidence intervals, are not typically applied to structural metrics generated by CiteSpace, which may constrain the interpretability and statistical robustness of the network analysis results. Additionally, the scope is limited to data from major databases like Web of Science, which may exclude relevant regional or emerging research. The focus on Western-centric perspectives may also reduce the applicability of findings in diverse global contexts. Furthermore, the exploration of AI in AMD diagnosis and treatment lacks a thorough examination of practical implementation barriers, and the long-term clinical outcomes of discussed therapies remain underexplored. Future research should address these gaps to provide a more comprehensive understanding of AMD.

This paper makes a substantial contribution to the field by providing an in-depth bibliometric analysis of recent trends, emerging research hotspots, and key contributors to AMD research over the past decade. By examining the evolution of literature, the role of various institutions and countries, and significant research trajectories, the study offers a comprehensive overview of the global landscape of AMD research. In addition, the investigation into the molecular mechanisms driving AMD pathogenesis—such as genetic influences, oxidative stress, complement–lipid interactions, and angiogenesis—further elucidates the disease’s progression and highlights potential therapeutic targets. The research also assesses preventive and therapeutic approaches, including pharmacological treatments, gene therapies, and nanomedicine, providing a holistic framework for clinical management. Looking ahead, the findings lay the groundwork for future research to optimize personalized treatment strategies, integrate artificial intelligence in diagnostic and therapeutic applications, and address challenges posed by comorbidities. The study also advocates for greater interdisciplinary collaboration to address the multifaceted nature of AMD, ultimately improving patient outcomes and offering new insights into long-term management and treatment approaches.

## Figures and Tables

**Figure 1 bioengineering-12-00548-f001:**
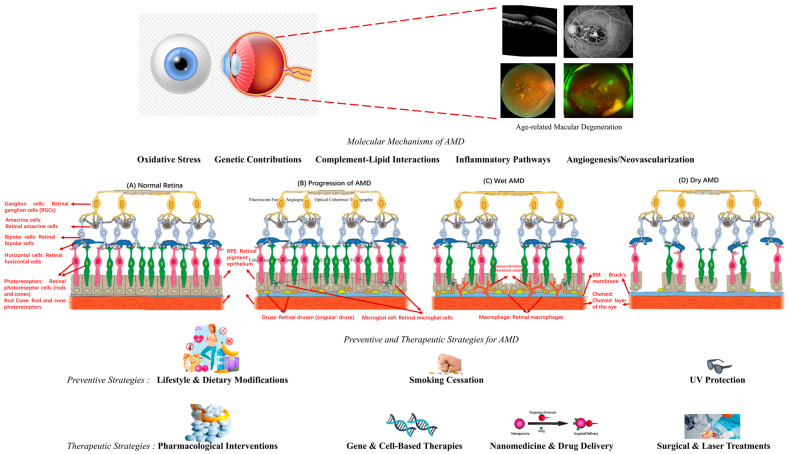
Molecular mechanisms underlie the pathogenesis of age-related macular degeneration.

**Figure 2 bioengineering-12-00548-f002:**
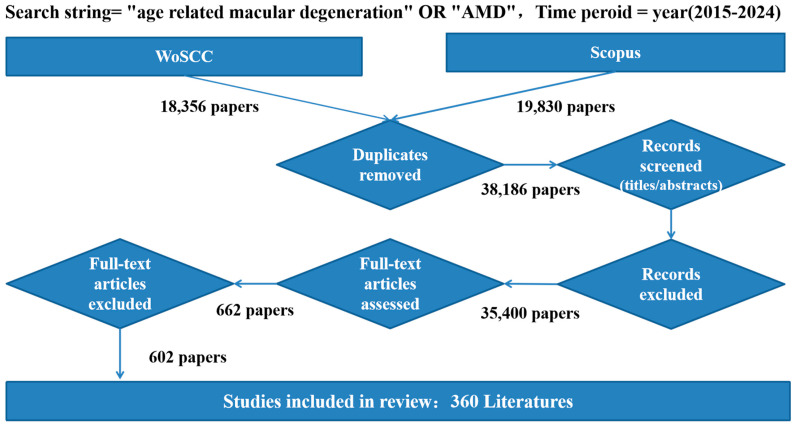
PRISMA flow diagram of the study selection process.

**Figure 3 bioengineering-12-00548-f003:**
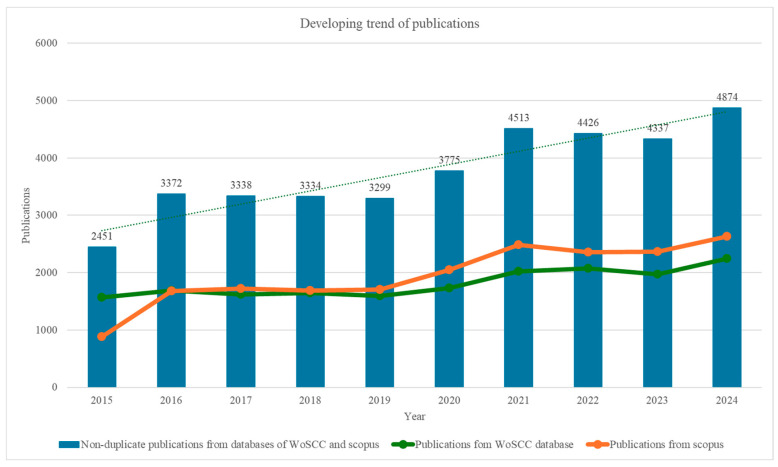
A historical trend analysis reveals changes in the number of publications across years.

**Figure 4 bioengineering-12-00548-f004:**
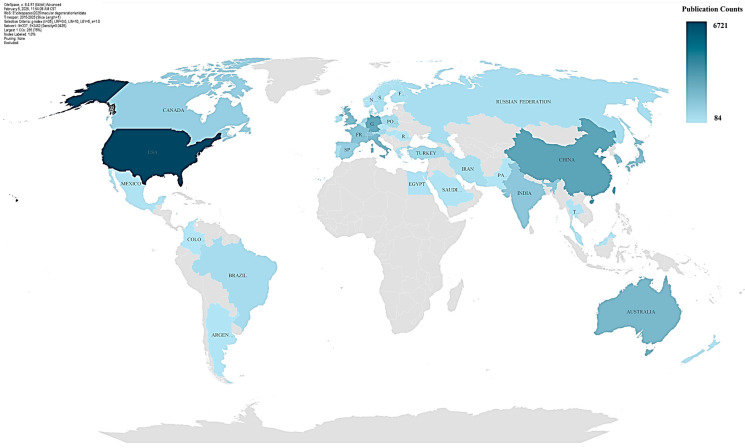
Countries are distributed based on their publication output.

**Figure 5 bioengineering-12-00548-f005:**
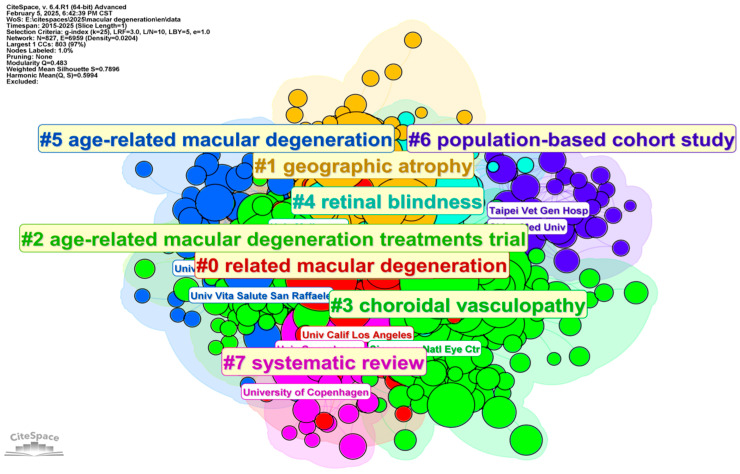
Institutions are distributed based on their publication frequency in the selected studies.

**Figure 6 bioengineering-12-00548-f006:**
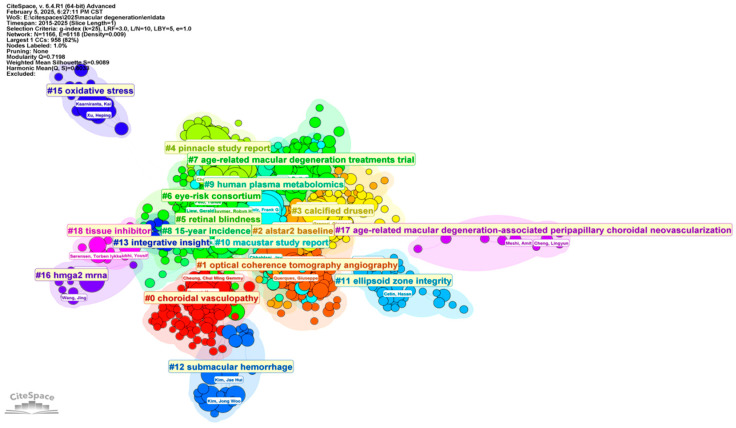
The distribution of cited authors is visualized.

**Figure 7 bioengineering-12-00548-f007:**
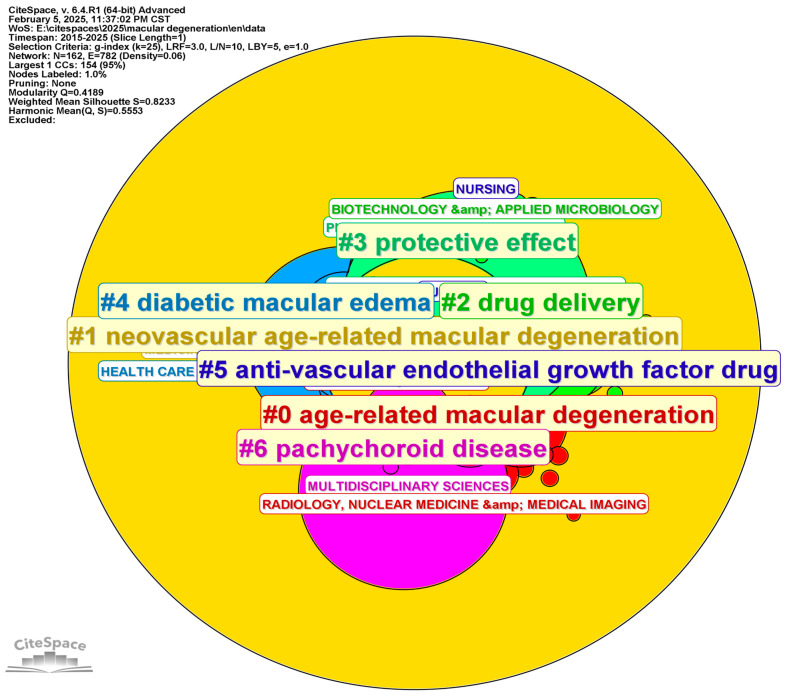
The distribution of categories is illustrated.

**Figure 8 bioengineering-12-00548-f008:**
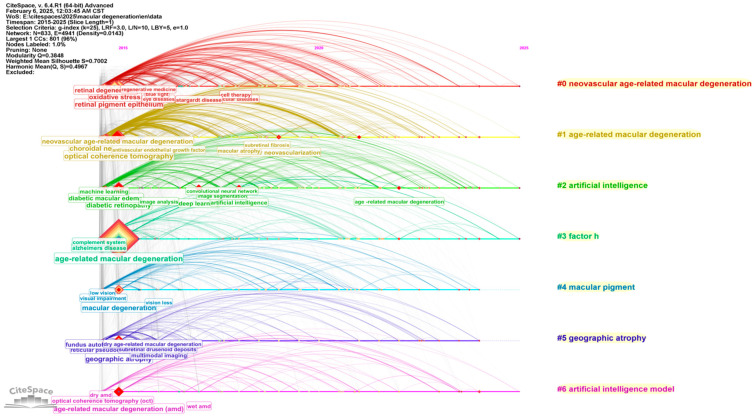
The developing history of keywords and clusters is illustrated.

**Figure 9 bioengineering-12-00548-f009:**
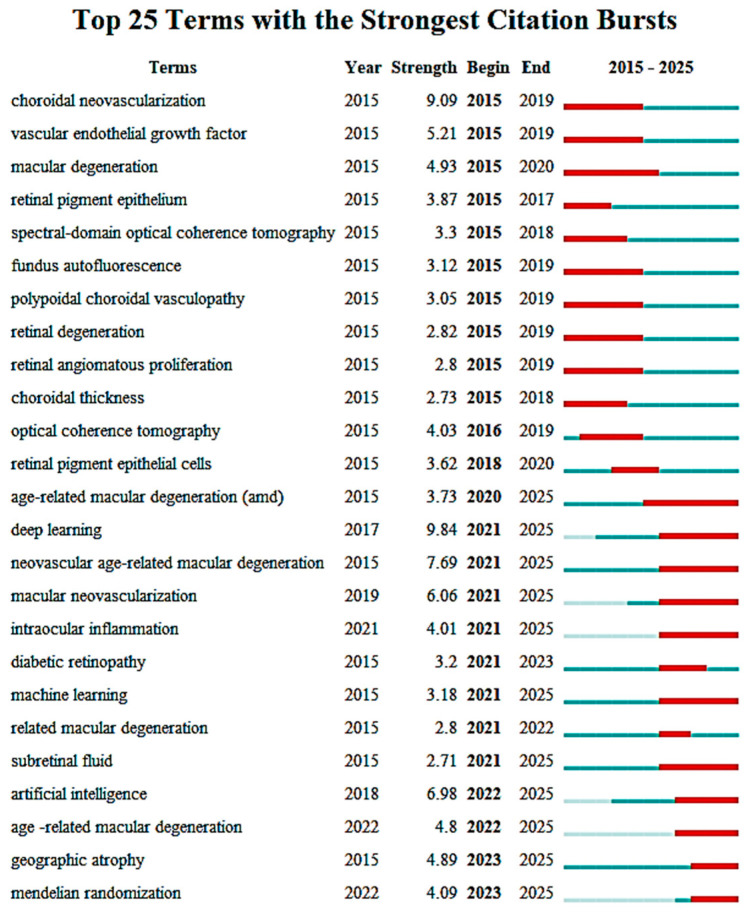
The top 25 citation burst history of keywords is illustrated.

**Figure 10 bioengineering-12-00548-f010:**
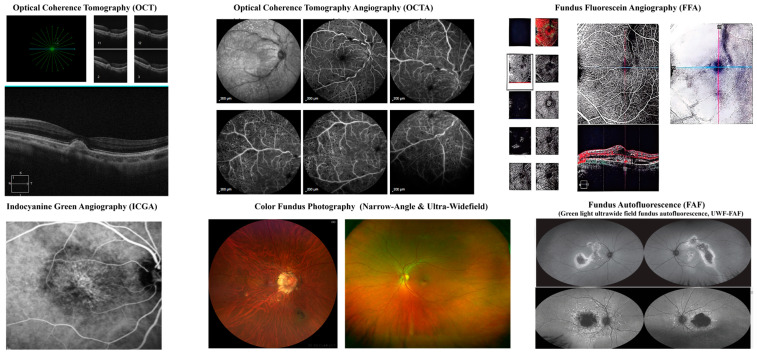
The medical imaging evidence of age-related macular degeneration is luustrated.

**Table 1 bioengineering-12-00548-t001:** Top 18 countries are listed according to the number of publications.

Ranking	Country	Publications Count	Centrality	Role in Research Network
1	USA	6721	0.03	Leading Contributor
2	United States	6458	0.08	Leading Contributor
3	China	2863	0.04	Emerging Leader
4	Germany	2716	0.06	Strong Collaborator
5	Peoples R China	2337	0.02	Emerging Leader
6	Australia	2009	0.04	Strong Contributor
7	Italy	2007	0.04	Strong Contributor
8	United Kingdom	1878	0.05	Global Hub
9	England	1667	0.02	Global Hub
10	Japan	1647	0.01	Regional Leader
11	South Korea	1629	0.02	Regional Leader
12	Switzerland	1555	0.06	Key Hub
13	France	1539	0.06	Key Hub
14	India	1390	0.03	Regional Contributor
15	Spain	1049	0.05	Key Collaborator
16	Canada	1047	0.11	Key Hub
17	Singapore	935	0.04	Key Collaborator
18	Netherlands	753	0.02	Moderate Contributor

**Table 2 bioengineering-12-00548-t002:** Top 12 institutions are listed according to the number of publications.

Ranking	Institutions	Publications Count	Centrality	Role in Research Network
1	University of Melbourne	430	0	Significant contributor to global research output, leading in impactful publications.
2	University of California	409	0	Prominent role in advancing interdisciplinary research within the network.
3	University of Sydney	392	0	Major contributor to research initiatives, particularly in collaboration with local and global institutions.
4	National Eye Institute	392	0	Key player in fostering innovation and collaboration in vision-related research.
5	University of California, Los Angeles	383	0	Leading institution in producing high-impact research, especially in healthcare and technology.
6	Harvard Medical School	349	0	Highly influential in medical research, facilitating widespread collaborations globally.
7	University College London (UCL)	300	0	Renowned for its pioneering research, with strong inter-institutional ties.
8	Duke University	289	0	Contributes significantly to collaborative research projects and knowledge dissemination.
9	National Institutes of Health (NIH)	285	0	Major hub for funding and facilitating groundbreaking biomedical research.
10	Moorfields Eye Hospital NHS Foundation	285	0	Key institution in ophthalmic research, connecting niche areas of expertise globally.
11	Sun Yat-Sen University	227	0	A prominent Asian institution showcasing increasing contributions to global research and collaboration.
12	Shanghai Jiao Tong University	193	0	Emerging as a significant research hub in Asia, contributing to advancements in interdisciplinary fields.

**Table 3 bioengineering-12-00548-t003:** Top 20 key journals are listed according to publication number.

Ranking	Journal Name	Record Count	Percentage of Total	Impact Factor (2023)
1	Investigative Ophthalmology & Visual Science (IOVS)	6137	18.43%	3.241
2	Retina: The Journal of Retinal and Vitreous Diseases	1129	3.39%	3.944
3	Ophthalmology	1033	3.10%	8.462
4	American Journal of Ophthalmology	817	2.45%	5.711
5	British Journal of Ophthalmology	721	2.17%	2.601
6	Graefe’s Archive for Clinical and Experimental Ophthalmology	700	2.10%	2.467
7	Eye	615	1.85%	2.806
8	Acta Ophthalmologica	604	1.81%	2.623
9	Experimental Eye Research	495	1.49%	3.291
10	PLOS ONE	466	1.40%	3.752
11	Archives of Ophthalmology	446	1.34%	5.201
12	Scientific Reports	429	1.29%	4.011
13	Clinical and Experimental Ophthalmology	347	1.04%	2.034
14	Clinical Ophthalmology	346	1.04%	1.952
15	European Journal of Ophthalmology	316	0.95%	2.098
16	Ophthalmologica	310	0.93%	1.365
17	International Journal of Molecular Sciences	308	0.93%	5.923
18	Ophthalmology Retina	304	0.91%	5.305
19	JAMA Ophthalmology	290	0.87%	7.492
20	Ophthalmologe	248	0.75%	1.452

**Table 6 bioengineering-12-00548-t006:** Top 19 categories of published studies are listed.

Ranking	Category	Publications Count	Centrality	Role in Research Network
1	Ophthalmology	10,170	0.07	Central
2	Biochemistry & Molecular Biology	1285	0.18	Central
3	Pharmacology & Pharmacy	995	0.15	Central
4	Multidisciplinary Sciences	919	0	Moderate
5	Medicine, Research & Experimental	903	0.19	Central
6	Medicine, General & Internal	859	0.04	Moderate
7	Cell Biology	843	0.07	Moderate
8	Chemistry, Multidisciplinary	460	0.04	Moderate
9	Neurosciences	443	0.16	Central
10	Genetics & Heredity	441	0.03	Moderate
11	Engineering, Biomedical	377	0.18	Central
12	Biotechnology & Applied Microbiology	292	0.15	Moderate
13	Geriatrics & Gerontology	290	0.08	Moderate
14	Immunology	273	0.07	Moderate
15	Chemistry, Medicinal	254	0.02	Moderate
16	Health Care Sciences & Services	237	0.02	Low
17	Radiology, Nuclear Medicine & Medical Imaging	233	0.08	Moderate
18	Nutrition & Dietetics	220	0.02	Low
19	Optics	195	0.09	Moderate
20	Food Science & Technology	195	0.04	Low

**Table 7 bioengineering-12-00548-t007:** Top 19 keywords are ranked according to the number of publications involved.

Ranking	Keyword	Count	Category	Insight
1	human	15,607	General Theme	Focus on human-centered studies
2	age-related macular degeneration	11,950	Disease Focus	Focus on age-related eye diseases
3	humans	11,594	General Theme	Human studies in general
4	article	10,165	General Theme	General term for academic articles
5	macular degeneration	10,097	Disease Focus	Focus on macular degeneration pathologies
6	male	7756	Demographic Focus	Demographic analysis
7	female	7479	Demographic Focus	Demographic analysis
8	aged	6800	Demographic Focus	Demographic analysis
9	optical coherence tomography	6578	Methodology	Use of advanced imaging techniques
10	controlled study	6238	Methodology	Reliance on structured methodologies
11	visual acuity	5391	Clinical Focus	Intersection of clinical treatments
12	retinal pigment epithelium	4750	Clinical Focus	Focus on retinal pathologies
13	ranibizumab	4316	Clinical Focus	Key treatment for AMD
14	diabetic retinopathy	3493	Clinical Focus	Focus on diabetic eye diseases
15	genetics	2272	Molecular Mechanism	Exploration of genetic factors
16	angiogenesis inhibitors	2687	Molecular Mechanism	Study of angiogenesis-related therapies
17	oxidative stress	2608	Molecular Mechanism	Study of oxidative stress mechanisms
18	deep learning	1940	Emerging Technology	Adoption of AI techniques
19	optical coherence tomography angiography	1129	Emerging Technology	Use of advanced imaging in ophthalmology

**Table 8 bioengineering-12-00548-t008:** AMD biomarkers are summarized based on imaging modality.

Imaging Modality	Key Biomarkers
OCT	Drusen, SRF, IRF, PED, Hyperreflective foci, Ellipsoid zone loss, GA, Fibrovascular scarring, Choroidal thickness changes [34]
OCTA	CNV, Capillary dropout, Choriocapillaris flow deficits, Dark halo sign, Vascular density changes [35]
FFA	Classic/Occult CNV, Window defects, Pooling in PED, Staining of fibrosis, Blocked fluorescence [36]
ICGA	Polypoidal lesions, Branching vascular networks, Hypofluorescence in GA, Late leakage from CNV [37]
Fundus Photography	Drusen, GA, Pigmentary changes, Subretinal hemorrhage, Exudates, Fibrotic scars [38]
FAF	Hyperautofluorescent drusen, Hypoautofluorescence in GA, Speckled autofluorescence, Ring of increased FAF [39]

**Table 9 bioengineering-12-00548-t009:** The prognosis of AMD based on imaging evidence is listed.

Image Type	Biomarker	Before Treatment (Signs of Active AMD)	After Treatment (Effective Response)
Optical Coherence Tomography (OCT)	Drusen	Large, confluent soft drusen	Reduction in drusen size or resorption
	Subretinal Fluid (SRF)	Hyporeflective spaces above the RPE	Decreased or complete resolution of SRF
	Intraretinal Fluid (IRF)	Cystoid spaces in the retina	Reduction or disappearance of IRF
	Retinal Pigment Epithelium Detachment (PED)	Elevated RPE with fluid accumulation	Flattening or decreased height of PED
	Hyperreflective Foci	Bright spots in retinal layers	Reduction in number or disappearance
	Ellipsoid Zone Disruption	Loss of photoreceptor integrity	Partial or full recovery of the ellipsoid zone
	Geographic Atrophy (GA)	Atrophic retinal areas with RPE loss	Stabilization (no further progression)
	Fibrovascular Scarring	Hyperreflective fibrotic tissue in the macula	No further growth of the scar
	Choroidal Thickness	Thinning (dry AMD) or thickening (wet AMD)	Stabilization or normalization
Optical Coherence Tomography Angiography (OCTA)	Choroidal Neovascularization (CNV)	Active CNV with abnormal vessels	Reduction in CNV size, decreased vascular density
	Capillary Dropout	Loss of retinal capillary network	Improved vascular perfusion or stability
	Choriocapillaris Flow Deficits	Non-perfused areas	Reduction in ischemic areas
	Dark Halo Sign	Hypointense ring around CNV	Decrease or disappearance
	Vascular Density Changes	Increased neovascularization or reduced capillary density	Stabilization or improvement in normal vasculature
Fundus Fluorescein Angiography (FFA)	Classic CNV	Well-defined hyperfluorescent lesion	Decrease or disappearance of leakage
	Occult CNV	Diffuse hyperfluorescence with leakage	Reduced fluorescence or stability
	Window Defects	Hyperfluorescence due to RPE atrophy	No expansion of defects
	Pooling in PED	Accumulation of dye in PED	Decreased pooling
	Staining of Fibrotic Scars	Persistent fluorescence without leakage	No further increase in staining
	Blocked Fluorescence	Hemorrhage or lipid exudates blocking fluorescence	Absorption of hemorrhage and reduced blockage
Indocyanine Green Angiography (ICGA)	Polypoidal Lesions	Active, round, hyperfluorescent structures	Decrease in lesion size or complete resolution
	Branching Vascular Networks	Abnormal choroidal vasculature	Reduction in complexity of networks
	Hypofluorescence in GA	Large areas of non-perfusion	No further enlargement of hypofluorescent zones
	Late Leakage from CNV	Persistent fluorescence in late phases	Decrease or stabilization
Color Fundus Photography (Ultra-Widefield & Narrow-Angle)	Drusen	Soft, large drusen in macula	Reduction in drusen size or disappearance
	Geographic Atrophy (GA)	Large atrophic regions	No further spread of GA
	Pigmentary Changes	RPE hyperpigmentation or hypopigmentation	Stabilization of pigmentary abnormalities
	Subretinal Hemorrhage	Red patches in wet AMD	Absorption of hemorrhage
	Exudates	Lipid deposits near the macula	Reduction in exudates
	Fibrotic Scars	White lesions due to fibrosis	No further growth of scar tissue
Fundus Autofluorescence (FAF)	Hyperautofluorescent Drusen	Increased metabolic activity in RPE	Reduced autofluorescence, indicating drusen regression
	Hypoautofluorescence in GA	Large areas of RPE loss	No expansion of atrophic regions
	Speckled Autofluorescence	Irregular FAF pattern, suggesting stressed RPE	Normalization or stability
	Ring of Increased FAF around GA	Predicts future atrophy expansion	Ring disappears or remains stable
	Patchy Hypoautofluorescence	Advanced disease with fibrosis	No worsening of patchy areas

**Table 10 bioengineering-12-00548-t010:** Key studies of AI applied to AMD are listed.

Studies	Data Source	Sample Size	Algorithm Architecture	Validation Methods	Performance Metrics
AMD detection [56]	OCT & FAF: Kaggle open-source datasets CFP (45°): Open-source + real-world data from Zhuhai People’s Hospital UWF (200°): In-house dataset from Shenzhen Aier Hospital	OCT: 32,347 images (AMD + normal) FAF: 1947 AMD + 2874 normal images CFP: 4445 AMD (445 open-source + 4000 real-world) 4874 normal (2000 open-source + 2874 real-world) UWF: 2300 images (1100 AMD + 1200 normal)	VGG16 and Improved VGG16 (VGG16 + Skip-attention)	Accuracy for the Unseen Testing Dataset Sensitivity Specificity AUC XAI Indicator Test Time/Second (Per Image)	Accuracy (Unseen Testing Dataset): Original VGG16: 57–97% (avg: 82%); Improved VGG16: 90.6–100% (avg: 96.62%) Sensitivity: Original VGG16: 61–94% (avg: 80%); Improved VGG16: 91–100% (avg: 96%) Specificity: Original VGG16: 46–95% (avg: 74%); Improved VGG16: 89.1–100% (avg: 96%) AUC (Area Under the Curve): Original VGG16: 80.88–91.01% (avg: 90.77%); Improved VGG16: 84.25–99.75% (avg: 94.58%) XAI Indicator (Explainability Score): Original VGG16: 0.3–1.0 (avg: 0.64); Improved VGG16: 0.6–1.0 (avg: 0.84) Test Time per Image (seconds): Original VGG16: 0.067–0.117 s (avg: 0.084 s); Improved VGG16: 0.07–0.45 s (avg: 18.50% increase overall)
The segmentations of the internal limiting membrane (ILM), retinal pigment epithelium (RPE), and RPE to Bruch’s membrane region [57]	Rear-word data from individuals participating in a prospective longitudinal study on AMD at the National Eye Institute, National Institutes of Health	Total OCT volumes: 402 (201 Spectralis for training, 201 Spectralis + 201 Cirrus for testing); Total B-scans: 25,728 (201 × 128)	U-Net and DeepLabV3	Mean absolute error (MAE) and mean squared error (MSE)	Mean Absolute Error (MAE): ILM: 1.82 ± 0.24 pixels (≈ 7.0 ± 0.9 μm); RPE: 2.46 ± 0.66 pixels (≈ 9.5 ± 2.6 μm) Dice Similarity Coefficient (RPEDC region): Dice: 0.87 ± 0.01
Classify changes in consecutive OCT B-scans; Predict 3-month structural evolution from a single time point [58]	MARIO Challenge dataset (MICCAI 2024) Includes OCT B-scans, infrared fundus localizer images, and clinical variables	Not explicitly stated, but based on official MARIO challenge dataset (participant-limited, multi-time-point OCT volumes)	Multimodal fusion deep learning framework; Feature extractors: Finetuned RETFound + EfficientNetV2; Input modalities: OCT B-scans + infrared fundus + clinical data	Evaluated on MARIO challenge benchmark tasks; Task-specific performance comparison using standard challenge metrics	F1 Score (Task 1—change detection): 0.851 F1 Score (Task 2–3-month prediction): 0.703
Evaluation of findable, accessible, interoperable, reusable (FAIR) principle compliance in public OCT datasets for AMD research [59]	Source Type: Public, open-access datasets Search Platforms: Google Dataset Search National Library of Medicine (NLM) Dataset Catalog PubMed Inclusion Criteria: AMD-specific research purpose Human-derived OCT images Open-access availability	Google Dataset Search: 42 datasets NLM Dataset Catalog: 142 datasets PubMed: 131 publications Final included datasets for analysis: 16 non-duplicate OCT datasets	None applied (focus was on dataset FAIR compliance assessment, not algorithmic modeling)	Evaluation against FAIR principles (Findable, Accessible, Interoperable, Reusable); Manual checklist-based analysis based on metadata and documentation review	Findable: 5%; Accessible: 82%; Interoperable: 73%; Reusable: 0%

## Data Availability

The sub-figures of OCT, OCTA, FFA, and Color Fundus Photography in Figure 10 are obtained from Zhuhai People’s Hospital (Ethics Approval: Beijing Institute of Technology Affiliated Zhuhai People’s Hospital Ethics Committee, Research Project Ethical Review Approval Document, Approval Number: [2025]-KT-21).

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
