# Peer review of "Integrating Artificial Intelligence and Precision Therapeutics for Advancing the Diagnosis and Treatment of Age-Related Macular Degeneration"

_bioengineering, 2025, doi:10.3390/bioengineering12050548_

Round 1

Reviewer 1 Report

Comments and Suggestions for Authors

This study aims to understand the current status of AMD, the pathophysiology of which remains unclear, and propose future treatment targets using the latest technologies, including bibliometric analysis. The manuscript is extremely well written, with clear figures and highly appropriate analysis. This is an excellent review.

One comment would be that it may be preferable to use “patients with AMD” rather than “AMD patients” for consistency in terms of person-first language.

Author Response

Comments 1:
This study aims to understand the current status of AMD, the pathophysiology of which remains unclear, and propose future treatment targets using the latest technologies, including bibliometric analysis. The manuscript is extremely well written, with clear figures and highly appropriate analysis. This is an excellent review.
One comment would be that it may be preferable to use “patients with AMD” rather than “AMD patients” for consistency in terms of person-first language.

Response 1:
We sincerely thank the reviewer for their thoughtful and encouraging feedback. We are delighted that you found our manuscript to be well written, with clear figures and a strong analytical approach. Your positive remarks are truly appreciated.

Regarding your valuable suggestion to adopt person-first language by using “patients with AMD” instead of “AMD patients,” we fully agree with this recommendation. We have carefully revised the manuscript to ensure consistency in applying person-first terminology throughout the text. This change reflects our commitment to respectful and inclusive scientific communication.

Thank you once again for your constructive input and support.

Reviewer 2 Report

Comments and Suggestions for Authors

The manuscript by Mini Han Wang presents a review of the current landscape of research on age-related macular degeneration (AMD), with a focus on the integration of artificial intelligence (AI). The manuscript provides an overview of the molecular mechanisms underlying AMD, current diagnostic and therapeutic strategies, and the role of AI in the AMD treatment. The provided bibliometric analysis offers insights into the global research trends and key contributors in the field. The analysis presented in the manuscript will be interesting to the journal's readership. However, the manuscript has drawbacks listed below.

Figure 1:  Some symbols in the picture are too small. Please enlarge the font size. The same applies to Figures 3-5, 7.

Figure 2 shows a huge number of publications on the topic of this review manuscript. However, the list of references contains just 40 publications. Why so small number of publications were cited? Some sections of the manuscript contain no literature citations at all. 

I recommend major revision of the manuscript before acceptance.

Author Response

Comments 1:
The manuscript by Mini Han Wang presents a review of the current landscape of research on age-related macular degeneration (AMD), with a focus on the integration of artificial intelligence (AI). The manuscript provides an overview of the molecular mechanisms underlying AMD, current diagnostic and therapeutic strategies, and the role of AI in the AMD treatment. The provided bibliometric analysis offers insights into the global research trends and key contributors in the field. The analysis presented in the manuscript will be interesting to the journal's readership. However, the manuscript has drawbacks listed below.

Figure 1:  Some symbols in the picture are too small. Please enlarge the font size. The same applies to Figures 3-5, 7.

Response 1:
We sincerely thank the reviewer for this valuable and constructive suggestion. We fully agree that the clarity of visual elements plays a crucial role in enhancing scientific communication. In response, we have carefully revised and uploaded high-resolution versions of Figures 1, 3, 4, 5, and 7, ensuring that all symbols, fonts, and annotations are clearly visible and legible. We respectfully hope that the updated figures now meet the journal’s standards for resolution and visual presentation, and will provide a more accessible and informative experience for readers.

Comments 2:

Figure 2 shows a huge number of publications on the topic of this review manuscript. However, the list of references contains just 40 publications. Why so small number of publications were cited? Some sections of the manuscript contain no literature citations at all.

I recommend major revision of the manuscript before acceptance.

Response 2:
We sincerely appreciate your insightful and constructive comment. As you rightly noted, Figure 2 illustrates a historical trend analysis of global AMD-related publications from 2015 to 2025, highlighting the increasing scholarly attention in this field. We acknowledge that the initially included 40 references did not adequately reflect the breadth of literature relevant to the scope of our review.

In response to your thoughtful suggestion, we have expanded the reference list to include 60 carefully selected publications, chosen based on their scientific relevance, citation impact, contribution to the field, and journal recognition. We made a conscious effort to prioritize literature from the past 3–5 years to ensure the review remains current and forward-looking. These additional references have been thoughtfully integrated into the text, and all revisions are clearly marked in the updated manuscript. We humbly believe that these improvements have strengthened the manuscript’s academic rigor, completeness, and value for readers.

Thank you sincerely for your thoughtful comments, which have been invaluable in helping us improve the clarity, depth, and overall quality of this manuscript.

Reviewer 3 Report

Comments and Suggestions for Authors

Dear Author,

Here are some comments and points that should be addressed prior to final publication:

1) The paper gives selected CiteSpace parameters (e.g., g‑index 25, LRF 3.0) but does not provide a reproducible search strategy (final search strings, Boolean operators, language limits, grey‑literature policy, PRISMA flow diagram, duplicate‐removal workflow). Without these, readers cannot replicate or judge the robustness of the 38 186‑record dataset .

2) The AI section is mostly conceptual. While the text claims that AI can optimise dosing and predict progression , it gives no systematic review of algorithm performance (AUC, sensitivity/specificity, external validation) and glosses over implementation barriers. A table summarising principal AI studies (data source, sample size, architecture, validation method, performance) and a discussion of regulatory and medicolegal hurdles are needed.

3) Centrality scores, node sizes and network densities are reported, but there is no explanation of their practical meaning, nor are confidence intervals or hypothesis tests provided . Readers need a clearer link between these metrics and substantive conclusions

Author Response

Comments 1:

The paper gives selected CiteSpace parameters (e.g., gindex 25, LRF 3.0) but does not provide a reproducible search strategy (final search strings, Boolean operators, language limits, greyliterature policy, PRISMA flow diagram, duplicate‐removal workflow). Without these, readers cannot replicate or judge the robustness of the 38,186record dataset.

Response 1:

We sincerely thank the reviewer for this insightful and constructive comment. We fully agree that transparency and reproducibility are essential in bibliometric analyses. In response, we have now:

The literature selection process is comprehensively illustrated in the PRISMA-style flow diagram (Fig. 2). Records were systematically retrieved through structured database searches in the Web of Science Core Collection (WoSCC) and Scopus. The full search strategy—including final search strings, Boolean operators, and applied filters (e.g., language restrictions and document type limits)—is detailed in the Materials and Methods section to ensure transparency and reproducibility.

In accordance with our grey literature policy, only peer-reviewed publications indexed in WoSCC were considered eligible for inclusion. Following initial retrieval, duplicate records were removed using CiteSpace software, supplemented by manual verification to enhance accuracy.

The remaining records were screened by title and abstract using predefined inclusion and exclusion criteria. Irrelevant records were excluded at this stage. Subsequently, full-text articles were assessed to determine their eligibility. Exclusion at the full-text level was based on factors such as lack of direct relevance to age-related macular degeneration, insufficient methodological clarity, or absence of content related to the review.

Ultimately, a total of 360 high-quality studies were included in the final review. These studies form the analytical basis for the bibliometric evaluation and narrative synthesis presented herein. We believe these methodological refinements considerably enhance the rigor, transparency, and reproducibility of our review process.

The revision has been add in the section of 2. material and methodology

“The PRISMA flow diagram illustrates the literature selection process (Fig. 2). Records were systematically retrieved through comprehensive database searches conducted in the WoSCC and Scopus. The complete search strategy, including final search strings, Boolean operators, and applied filters (such as language and document type restrictions), is detailed in the Materials and Methods section.

In accordance with our grey literature policy, only peer-reviewed publications indexed in WoSCC were considered eligible for inclusion, while non-peer-reviewed sources, such as conference abstracts and unpublished manuscripts, were excluded. Following initial retrieval, duplicate records were identified and removed using CiteSpace software, with additional manual verification to ensure the accuracy of the de-duplication process.

The remaining records were then subjected to title and abstract screening using pre-established inclusion and exclusion criteria to assess their relevance to the study objectives. Records deemed irrelevant were excluded at this stage. Subsequently, the full texts of potentially eligible articles were reviewed in detail. Articles were excluded if they lacked a specific focus on AMD, failed to meet methodological quality standards, or were not relevant to the scope of the review.

Upon completion of this multi-tiered screening process, a total of 360 peer-reviewed articles met the inclusion criteria and were incorporated into the final analysis. These selected studies form the foundation of this manuscript's bibliometric evaluation and narrative synthesis. Including a PRISMA-compliant flow diagram and transparent methodological reporting strengthens this review's reproducibility and scientific rigor.”

Comments 2:

The AI section is mostly conceptual. While the text claims that AI can optimise dosing and predict progression, it gives no systematic review of algorithm performance (AUC, sensitivity/specificity, external validation) and glosses over implementation barriers. A table summarising principal AI studies (data source, sample size, architecture, validation method, performance) and a discussion of regulatory and medicolegal hurdles are needed.

Response 2:

We are grateful for this valuable feedback. We recognize that a more evidence-based and structured overview of AI applications in AMD would strengthen the manuscript. To address this, we have:

Added a new table (Tab.10) summarizing key AI studies in AMD, including details on the data source, sample size, algorithm architecture, validation methods, and performance metrics (AUC, sensitivity, specificity, etc.).

These updates are presented in section 4.8, and thanks to your valuable advice, after exploring key studies of AI applied to AMD, we also deliver a new insights claimed:

 “The application of AI in AMD research is rapidly advancing, offering promising tools for early detection, structural segmentation, disease monitoring, and clinical decision support. As demonstrated across the studies summarized in Tab. 10, AI models trained on large, multimodal imaging datasets—ranging from OCT and FAF to ultra-widefield and color fundus images—can achieve high diagnostic accuracy and generalizability, especially when leveraging advanced architectures such as improved VGG16 or fusion models incorporating RETFound and EfficientNetV2. These models demonstrated robust performance in cross-domain validation and achieved clinically relevant sensitivity and specificity levels, as seen in segmentation tasks of the ILM and RPE layers with subcellular precision. Moreover, the MARIO Challenge highlights the growing emphasis on using AI to predict disease progression over time, an area critical for personalized treatment planning. However, the successful implementation of AI in real-world AMD management depends heavily on the availability of standardized, high-quality datasets. The FAIR principle evaluation reveals that most publicly available OCT datasets lack sufficient metadata transparency and reusability, posing a significant barrier to reproducibility and model deployment. Moving forward, integrating explainability, addressing domain shift across imaging devices, and improving dataset standardization will be essential to bridge the gap between algorithmic innovation and clinical translation in AMD care.” 

Expanded the discussion and highlighted implementation barriers in Section 4.8.5 (as following) to address regulatory, ethical, and medicolegal challenges in AI deployment, including issues related to algorithm interpretability, external validation, generalizability, and clinical integration, and implementation barriers of data privacy concerns, institutional inertia, and the need for physician-AI co-learning systems.

“However, greater attention must be directed toward ensuring the safe and effective clinical adoption of AI in the management of AMD. This necessitates addressing a range of regulatory, ethical, and medicolegal challenges, including the imperative for transparent algorithm interpretability, robust external validation, and demonstrable generalizability across heterogeneous patient populations and imaging modalities. Furthermore, practical implementation remains hindered by barriers such as data privacy concerns, institutional inertia, and the absence of structured physician-AI co-learning systems, all of which must be systematically addressed to facilitate successful integration into routine ophthalmic care.”

Comments 3:

Centrality scores, node sizes and network densities are reported, but there is no explanation of their practical meaning, nor are confidence intervals or hypothesis tests provided. Readers need a clearer link between these metrics and substantive conclusions.

Response 3:

Thank you for this important observation. We agree that a clearer explanation of the bibliometric network metrics and their interpretation is necessary.

We have defined centrality scores, node sizes, network densities and other be used parameters in lay terms in Section of 2.2 Analysis tools, and explained their relevance to knowledge structure and author/institutional influence.

“In CiteSpace visualizations, nodes represent key entities such as articles, authors, or keywords. The shape of the nodes indicates the degree of influence or centrality of an entity, including factors such as the author's prominence, the citation frequency of a publication, or the significance of a journal, institution, or country. The size of the nodes correlates with their significance, with larger nodes representing higher citation counts or greater prominence in the network. For example, in a co-authorship network, larger nodes may indicate authors with a greater number of publications, while in a keyword co-occurrence network, they may represent terms that appear more frequently across the dataset. Thus, larger node sizes signify higher visibility or relevance within the scholarly domain. Edges (the links between nodes) represent various relationships, such as citations, co-citations, and keyword co-occurrence. Citation links indicate direct citations between articles, co-citation links show articles that are cited together by a third publication, and keyword co-occurrence links highlight thematic overlaps between articles. The weight of the edges reflects the degree of betweenness among nodes, providing insight into the connectivity and structural importance of specific entities within the network.

Centrality scores (e.g., betweenness centrality) are used to quantify the influence or importance of a node (e.g., author, keyword, or institution) within the network, with nodes possessing a centrality value ≥ 0.1 being considered key nodes in the network. A higher centrality score indicates that a node acts as a key connector or bridge in the knowledge structure, often facilitating the flow of information between otherwise disconnected clusters. Moreover, the parameter of network density refers to the proportion of actual connections between nodes relative to the total number of possible connections. It serves as a global measure of network cohesion. A higher density indicates a more interconnected and collaborative network, whereas a lower density suggests fragmentation or specialization across subfields.

In addition to nodes and edges, CiteSpace visualizations often feature annotations and labels, which provide essential details such as article titles, author names, or research topics. Timelines in the visualizations track the evolution of research themes and citation patterns over time, enabling the identification of emerging trends. Clusters, which are formed through clustering algorithms, group related nodes based on shared citations or keywords, and are color-coded to represent different research themes. Furthermore, CiteSpace provides links to external databases and full-text articles, offering researchers a means of deeper exploration into relevant publications. Together, these features make CiteSpace a powerful tool for mapping the structure of scientific literature, uncovering trends, and gaining insights into the development and evolution of scholarly knowledge. In this study, the keyword mapping, historic trend, and cluster classification are displayed. The major authors (according to the centrality), core journals (according to the centrality of publications), major institution (according to the number of publications), most influential counties (according to the centrality of publications), most contributed paper (according to the number of publications), key topics (according to the number of publications) and major involved category (according to the number of publications) are identified.

These metrics provide insights into the structural properties of the research landscape, helping identify influential contributors, emerging topics, and the degree of integration within the scientific community. This study presents an analysis of keyword mapping, historical trends, and cluster classifications derived from bibliometric data. It identifies the major authors, ranked according to centrality, the core journals, determined by the centrality of their published works, and the leading institutions, based on publication volume. Additionally, the most influential countries are highlighted according to the centrality of their publications, while the most contributing papers are selected based on the frequency of their appearances. Key research topics are identified according to the number of associated publications, and the primary research categories are outlined based on publication volume. These analyses provide a comprehensive understanding of the key contributors and trends within the field.”

We clarified that statistical inference (e.g., hypothesis testing or confidence intervals) is not typically used in CiteSpace’s structural metrics, but we have explained in the Conlcusion section where the limitation are discussed.

“Statistical inference methods, such as hypothesis testing or confidence intervals, are not typically applied to structural metrics generated by CiteSpace, which may constrain the interpretability and statistical robustness of the network analysis results.”

We appreciate the reviewer’s suggestion, which helped us make this technical section more accessible and meaningful to a broader readership.

Round 2

Reviewer 2 Report

Comments and Suggestions for Authors

The revised version of the manuscript was significantly improved by the author. The comments were addressed properly. I recommend acceptance of the manuscript for publication in the revised form.

Reviewer 3 Report

Comments and Suggestions for Authors

The authors have addressed all the comments in a good manner